# Design Principles for Sequence Models via Coefficient Dynamics

## Abstract

Deep sequence models, ranging from Transformers and State Space Models (SSMs) to more recent approaches such as gated linear RNNs, fundamentally compute outputs as linear combinations of past value vectors. To draw insights and systematically compare such architectures, we develop a unified framework that makes this output operation explicit, by casting the linear combination coefficients as the outputs of autonomous linear dynamical systems driven by *impulse inputs*. This viewpoint, in spirit substantially different from approaches focusing on connecting linear RNNs with linear attention, reveals a common mathematical theme across diverse architectures and crucially captures softmax attention, on top of RNNs, SSMs, and related models. In contrast to new model proposals that are commonly evaluated on benchmarks, we derive design principles linking architectural choices to model properties. Thereby identifying tradeoffs between expressivity and efficient implementation, geometric constraints on input selectivity, and stability conditions for numerically stable training and information retention. By connecting several insights and observations from recent literature, the framework both explains empirical successes of recent designs and provides guiding principles for systematically designing new sequence model architectures.

## 1 Introduction

Sequence modeling lies at the core of modern machine learning, powering advances in natural language processing, computer vision, speech recognition, and robotics. At the heart of this field is a deceptively simple question (Elman, 1990): how should a model combine past information to produce meaningful representations and make predictions about the future? Classical approaches such as recurrent neural networks (RNNs; Hochreiter & Schmidhuber (1997)) framed this as a recursive state-update problem, whereas the Transformer (Vaswani et al., 2017) introduced attention mechanisms that compute adaptive linear combinations over past tokens. More recently, state space models (SSMs) (Gu et al., 2022a; Gu & Dao, 2023) or equivalently linear attention models (Katharopoulos et al., 2020; Dao & Gu, 2024; Schlag et al., 2021) have further broadened the design space by importing ideas from dynamical systems, delivering foundation models with linear sequence length complexity. Despite their differences in formulation, computational complexity, and performance on crucial benchmark tasks (Arora et al., 2023; Jelassi et al., 2024), both linear and softmax attention models share a common thread: their core sequence mixing mechanism computes a linear combination of value vectors with coefficients determined by the interaction of queries, keys, and additional learned parameters. For attention, these coefficients are derived through similarity functions and normalizations; for SSMs and (gated) linear attention, they emerge from the evolution of hidden states under linear dynamics. Yet, while this unifying perspective is informally recognized, most architectural innovations are still introduced and validated primarily through benchmarks. This benchmark-driven approach has fueled rapid empirical progress, but complicates the disentanglement of which design elements – e.g. normalization, gating, or state-update dynamics – truly account for observed improvements. What is lacking is a set of general design principles that explain why specific models succeed and guide the systematic development of new ones, beyond approaches focusing on *ad-hoc* analyses of single-task setups (Jelassi et al., 2024; Merrill et al., 2024; Arora et al., 2024).

In this work, we develop a principled framework for sequence model design grounded in dynamical systems theory. Specifically, we show that the coefficients governing linear combinations of value vectors, can be expressed as the output of autonomous linear dynamical systems subject to *impulse*

inputs. Our approach reveals the common mathematical structure underlying diverse architectures and provides a principled foundation for designing new sequence models. Since the present study is restricted to single-layer models, multi-layer composition can produce richer behaviors – such as formation of induction heads (Olsson et al., 2022) – beyond what our base framework captures. Nevertheless, by laying out the core design principles at the layer level, we aim to establish the foundations for understanding and designing more complex systems. Our *contributions* are as follows:

- **Design principles** for sequence models that stem from a unifying theory on the dynamic computations in sequence models, clarifying the role of sequence model components.
- **Analysis of tradeoffs and constraints** concerning expressivity, efficient implementation, input selectivity, and stability requirements in sequence models.
- **A unifying formalization of sequence models** that captures most recent architectures as special cases (Table 1), enabling systematic comparison without reliance on experimental benchmarks.
- **Empirical validation** demonstrates that our theory translates into practice.

## 2  RELATED WORK

We have recently witnessed a revived interest in efficient RNNs and linear attention mechanisms in language modeling as well as image/video processing (Liu et al., 2024) and DNA modeling (Nguyen et al., 2023; Schiff et al., 2024). Stemming from the seminal S4 work by Gu et al. (2020; 2022a), early SSMs (Gu et al., 2022b; Smith et al., 2023) and linear recurrent networks (Orvieto et al., 2023), outperformed attention on pattern-recognition and long-range reasoning benchmarks (Tay et al., 2021) and were rapidly adapted to the language domain (Wang et al., 2022; Fu et al., 2023). RetNet (Sun et al., 2023) and GateLoop (Katsch, 2023) paved the way for Mamba (Gu & Dao, 2023), which combined input selectivity with efficient training (linear in sequence length) to achieve improved performance in language modeling. Drawing inspiration from the known connection between RNNs and linear attention (Katharopoulos et al., 2020; Schlag et al., 2021), advancements such as GLA, Deltanet, xLSTM (Yang et al., 2023; 2024; Beck et al., 2024), and Mamba2 (Dao & Gu, 2024) have aided the unification of modern efficient sequence modeling approaches in terms of latent efficient matrix multiplications (Yang & Zhang, 2024). On top of this unified framework – *which crucially does not contain softmax attention* – several improvements were recently proposed, such as Gated Deltanet (Yang et al., 2025a), DeltaProduct (Siems et al., 2025), and Log-Linear Attention (Guo et al., 2025). We have also witnessed cross-fertilization between new linear RNN/attention ideas (forget gate, positional encodings) and the softmax world, with examples such as FoX (Lin et al., 2025) and Path (Yang et al., 2025b).

As flagship models are starting to adopt hybrid solutions based on mixing linear and softmax attention to enhance training and inference efficiency (AI21 Labs, 2024; Nano, 2025; Qwen Team, 2025), additional work is needed to further understand which architectural properties impact performance depending, on the specific nature (e.g. recall, associativity, memorization, compression) of the task at hand (Poli et al., 2024). While early studies show promise across benchmarks, especially for hybrid models at high pretraining budgets (Waleffe et al., 2024), others point to fundamental limitations of fixed-memory processing in recall, copy (Arora et al., 2023; Jelassi et al., 2024) and retrieval tasks (Guo et al., 2025). Expressivity of new linear RNN layers is well-studied (Orvieto et al., 2024; Cirone et al., 2024; Merrill et al., 2024), yet less is known about how architectural details impact model capabilities – especially as linear RNNs can be substantially more challenging to optimize (Zucchet & Orvieto, 2024; Okpekpe & Orvieto, 2025), – and we lack a unified framework able to capture both mechanisms under the same formalism, without resorting to infinite-dimensional expansions (Sieber et al., 2024). Our *coefficient dynamics* are inspired by this issue.

## 3  LINEAR COMBINATIONS WITH TEMPORAL DYNAMICS

We consider a one-layer causal sequence model (e.g. attention, S6), i.e. a parametric mechanism that defines $\{x_j\}_{j=1}^i \mapsto y_i$, where $i$ denotes the (time) position index $i = 1, \ldots, L$ and $x_i \in \mathbb{R}^d, y_i \in \mathbb{R}^{d_v}$ are the input and output respectively. In the following, we provide a formulation that encompasses most existing sequence model architectures using what we name *coefficient dynamics*. Following the standard transformer notation (Vaswani et al., 2017), and given the duality between attention, SSMs, and other recurrent architectures (Dao & Gu, 2024; Sieber et al., 2024), we define:

$$q_i = W_Q x_i, \quad k_i = W_K x_i, \quad v_i = W_V x_i, \tag{1}$$

where $W_Q \in \mathbb{R}^{n \times d}$, $W_K \in \mathbb{R}^{n \times d}$, and $W_V \in \mathbb{R}^{d_v \times d}$ are the learned parameters of the model.

Let $\{v_j\}_{j=1}^{i}$ denote the value vectors at (time) position $i$, then most existing sequence model can be fundamentally viewed as computing linear combinations of these value vectors:[1]

$$y_i = \sum_{j=1}^{i} \frac{\alpha_{i,j}}{\eta_i} v_j, \tag{2}$$

where each coefficient $\alpha_{i,j}$ represents the contribution of the value vector at position $j$ to the output at position $i$ and $\eta_i : \mathbb{R} \to \mathbb{R}$ denotes the normalization factor. In standard architectural proposals, such as softmax attention or linear attention, coefficients $\alpha_{i,j}$ are typically computed using queries, keys, and additional learned parameters. For consistency with existing literature, we refer to $\alpha_{i,j}$ as the *attention coefficients*.

In our framework, we interpret the coefficients $\alpha_{i,j}$ as the output measurement of a latent linear dynamical system with impulse input, which we call the *coefficient dynamics*. In contrast to previous works, which study how tokens are mixed as the iteration counts progress (Ali et al., 2024), we consider the tokens individually. This orthogonal perspective allows to precisely model both softmax and linear attention, with no need for infinite dimensional approximations. Specifically, for each key position $j$, we define an autonomous dynamical system with initial state $h_{i-1,j} = 0, \forall i \leq j$, i.e., where in standard dynamical systems' terminology, $h_{i,j} \in \mathbb{R}^n$, $u_i \in \mathbb{R}^n$, and $\alpha_{i,j} \in \mathbb{R}$ are the

$$h_{i,j} = A_i h_{i-1,j} + b_i u_i, \tag{3a}$$

$$u_i = \begin{cases} k_j, & \text{if } i = j, \\ 0 & \text{otherwise}, \end{cases} \tag{3b}$$

$$\alpha_{i,j} = \phi\left(q_i^\top h_{i,j}\right), \tag{3c}$$

state, input, and output of the system at time $i$ with an impulse input at time $j$, respectively. The readout map is given by $g_i(x) := \phi\left(q_i^\top x\right)$, where $q_i \in \mathbb{R}^n$ is the query, $k_j \in \mathbb{R}^n$ is the key, and $\phi(\cdot) : \mathbb{R} \to \mathbb{R}$ is a general nonlinear map. $A_i \in \mathbb{R}^{n \times n}$ and $b_i \in \mathbb{R}$ are the evolution matrix and the input scaling, respectively. For $A_i$ we focus on scalar matrices, $A_i = \lambda_i \mathbb{I}_n$, diagonal matrices, $A_i = \text{diag}(\lambda_i)$, and Householder matrices.[2] However, generally $A_i$ can have no particular structure. We consider the *autonomous* evolution of dynamical system equation 3, i.e., given a zero initial condition, we study the system's evolution when it receives an impulse input at time $j$ and then evolves without further input.

**Remark 1.** *This formulation differs from existing state space models (e.g., Mamba) in two key aspects: (i) the system receives input only at a single time step $i = j$, and (ii) the hidden state is zero immediately before the impulse, not at global time $i = -1$.*

It is possible to recover an analytic expression for the coefficients $\alpha_{i,j}$ in equation 2 by writing the convolution expression associated with equation 3. In particular, the state (or "transformed key") evolves, for $j \leq i$, as $h_{i,j} = \left(\prod_{t=j+1}^{i} A_t\right) b_j k_j$. Hence, we obtain the form:

$$\alpha_{i,j} = \phi(x_i^\top T_q^\top T_{h,i} x_j), \qquad \text{with} \quad T_{h,i} := \left(\prod_{t=j+1}^{i} A_t\right) b_j W_K, \quad T_q := W_Q. \tag{4}$$

This interpretation allows for the identification of four classes of parameters with distinct roles:

- **Readout map ($\phi(\cdot)$):** Applies pointwise (often nonlinear) transformations to dot product $q_i^\top h_{i,j}$.
- **Evolution matrices ($A_t$):** Control the temporal evolution of keys through scaling (diagonal matrices) or rotation (orthogonal matrices). Importantly, the evolution matrices define a sequence of transformation on the keys between input time $j$ and readout time $i$.
- **Scaling parameters ($b_j$):** Scale individual keys at their input time $j$.
- **Normalization factors ($\eta_i$):** Normalize the $\alpha_{i,j}$ coefficients, before the linear combination.

---

[1] For simplicity, we ignore potential output projections from the linear combination to the output.

[2] A *Householder matrix* is an orthogonal matrix of the form $A_i = I - 2z_i z_i^\top / \|z_i\|^2$, with $z_i$ a nonzero vector; and represents a reflection through the hyperplane orthogonal to $v_i$.

This framework provides a unified lens for understanding existing architectural proposals. We show in Table 1 that popular architectures can be recovered as special cases of this framework.

Table 1: Coefficient dynamics of popular architectures; the derivations are provided in Appendix B.

| Architecture | Parameters | | | | Coefficients |
|---|---|---|---|---|---|
| | $A_t$ | $b_j$ | $\phi(\cdot)$ | $\eta_i$ | $\alpha_{i,j}$ |
| Softmax Attn. (Vaswani et al., 2017) | $\mathbb{I}_n$ | $\frac{1}{\sqrt{n}}$ | $\exp(\cdot)$ | $\sum_{j=0}^{i}\alpha_{i,j}$ | $\frac{\exp(q_i^\top k_j/\sqrt{n})}{\sum_{j=1}^{i}\exp(q_i^\top k_j/\sqrt{n})}$ |
| Linear Attn. (Katharopoulos et al., 2020) | $\mathbb{I}_n$ | $\frac{1}{\sqrt{n}}$ | $\psi(\cdot)\psi(\cdot)$ | $\sum_{j=0}^{i}\alpha_{i,j}$ | $\frac{\psi(q_i)^\top\psi(k_j/\sqrt{n})}{\psi(q_i)^\top\sum_{j=1}^{i}\psi(k_j/\sqrt{n})}$ |
| Normalized Attn. (Sieber et al., 2024) | $\mathbb{I}_n$ | $\frac{1}{\sqrt{n}}$ | $\mathrm{Id}(\cdot)$ | $\eta_i$ | $\frac{q_i^\top k_j}{\eta_i\sqrt{n}}$ |
| GLA (Yang et al., 2023) | $\mathrm{diag}(\alpha_i)$ | $\frac{1}{\sqrt{n}}$ | $\mathrm{Id}(\cdot)$ | $1$ | $q_i^\top\left(\prod_{t=j+1}^{i}\mathrm{diag}(\alpha_t)\right)\frac{k_j}{\sqrt{n}}$ |
| Mamba-2 (Dao & Gu, 2024) | $e^{-\Delta_t A}$ | $\Delta_j$ | $\mathrm{Id}(\cdot)$ | $1$ | $q_i^\top\left(\prod_{t=j+1}^{i}e^{(-\Delta_t A)}\right)\Delta_j k_j$ |
| DeltaNet (Schlag et al., 2021) | $\mathbb{I}-\beta_t k_t k_t^\top$ | $\frac{\beta_j}{\sqrt{n}}$ | $\mathrm{Id}(\cdot)$ | $1$ | $q_i^\top\left(\prod_{t=j+1}^{i}\mathbb{I}-\beta_t k_t k_t^\top\right)\frac{\beta_j k_j}{\sqrt{n}}$ |
| Gated DeltaNet (Yang et al., 2025a) | $\alpha_t(\mathbb{I}-\beta_t k_t k_t^\top)$ | $\frac{\beta_j}{\sqrt{n}}$ | $\mathrm{Id}(\cdot)$ | $1$ | $q_i^\top\left(\prod_{t=j+1}^{i}\alpha_t(\mathbb{I}-\beta_t k_t k_t^\top)\right)\frac{\beta_j k_j}{\sqrt{n}}$ |
| mLSTM (Beck et al., 2024) | $e^{f_t}$ | $\frac{e^{i_j}}{\sqrt{n}}$ | $\mathrm{Id}(\cdot)$ | $\frac{\eta_i}{o_i}$ | $\frac{o_i}{\eta_i}q_i^\top\left(\prod_{t=j+1}^{i}e^{f_t}\right)\frac{e^{i_j}}{\sqrt{n}}k_j$ |

## 4 DESIGN PRINCIPLES FOR SEQUENCE MODELS

In the following, we provide principles that can inform the architecture design for sequence models. These principles are grounded in theoretical results for the coefficient dynamics (as described in equation 2 & equation 3) and followed by a discussion of their practical applicability. Formal proofs for the lemmas and corollaries are provided in Appendices C.1 and C.2, respectively.

### 4.1 ON CHOOSING THE READOUT MAP $\phi(\cdot)$

A fundamental issue with softmax attention is its quadratic complexity in sequence length, directly linked to its nonlinear readout map $\phi(\cdot) = \exp(\cdot)$. This fact is captured by the following principle.

**Principle 1.** *On modern hardware, a sequence model can only be efficiently computed using a recurrent formulation (e.g. parallel scan) if the readout function $\phi(\cdot)$ is linear.*

**Lemma 1.** *A recurrent formulation of equation 3 with finite memory (state) in $\mathbb{R}^{n \times d_v}$, which allows simultaneous computation of $\alpha_{i,j}$, exists if and only if $\phi(\cdot) : \mathbb{R} \to \mathbb{R}$ is a linear map.*

Various linear attention (e.g., Katharopoulos et al. (2020); Choromanski et al. (2021)) and SSM/RNN (e.g., Dao & Gu (2024); Beck et al. (2024); De et al. (2024)) proposals implicitly utilize this principle to design sequence models with linear complexity in sequence length. However, choosing a linear readout map $\phi$ has several geometric implications, as discussed below, which directly affect performance on crucial recall and retrieval tasks (Arora et al., 2024; Guo et al., 2025).

Beyond implementation considerations, in domains such as language processing, an important feature of sequence models is their ability to selectively attend to different tokens in the input sequence. This is often referred to as *input selectivity*. Mathematically, this translates to the model's capacity to set coefficients $\alpha_{i,j}$ equal to zero (or near-zero in practice) for uninformative tokens and to high values for informative tokens. Crucially, how informative a token is depends on the query, as is clear for tasks such as associative recall (Arora et al., 2023).

**Principle 2.** *The capability of a model to robustly set coefficients to zero (or near-zero) depends on the geometry of the zero-level set of the readout map $\phi(\cdot)$. Intuitively, if the zero (or near-zero) set has large measure in $\mathbb{R}$, values can be suppressed more robustly. Conversely, if the zero set is thin, suppression is fragile to learn.*

**Lemma 2.** *Let $\phi : \mathbb{R} \to \mathbb{R}$ be the readout map in equation 3 with connected zero-level set $\mathcal{Z} = \{z \in \mathbb{R} \mid \phi(z) = 0\}$, and let $z = q_i^\top h_{i,j}$, with $T_q, T_{h,i}$ defined in equation 4. Let $\mathcal{T}$ be the set of linear transformations that achieve the zero-level set, i.e., $\mathcal{T} = \{(T_q, T_{h,i}) \mid q_i^\top h_{i,j} \in \mathcal{Z} \text{ s.t. } q_i = T_q x_i, h_{i,j} = T_{h,i} x_j\}$ for any pair of inputs $x_i, x_j \in \mathbb{R}^d$. Then, the measure of $\mathcal{T}$ is directly proportional to the measure of the zero-level set $\mathcal{Z}$, i.e., $|\mathcal{T}| = c|\mathcal{Z}|$ with $c > 0$.*

Although Lemma 2 only considers zero-level sets, large near-zero level sets $\{x \mid -\epsilon \leq \phi(x) \leq \epsilon\}$ with small $\epsilon > 0$, provide effective input selectivity in practice. Intuitively, the proof follows from the fact that for an interval $\mathcal{Z}$, we can construct a set of linear combinations $\mathcal{T}_z = \{(T_q, T_{h,i}) \mid q_i^\top h_{i,j} = z\}$ that achieves $z$ for each $z \in \mathcal{Z}$. A larger set $\mathcal{Z}$ then implies that $\mathcal{T}$ is constructed from more subsets $\mathcal{T}_z$. Hence, when the zero-level set $\mathcal{Z}$ has large measure in $\mathbb{R}$ ($|\mathcal{Z}| \gg 0$), many linear transformations $(T_q, T_{h,i})$ can achieve $\alpha_{i,j} = 0$, enabling robust input selectivity. While our analysis establishes the existence of these beneficial configurations, determining whether gradient descent actually converges to any transformation in $\mathcal{T}$, requires a deeper analysis of specific optimization dynamics, loss functions, and architectures. However, as argued in Goodfellow et al. (2016), the abundance of high-dimensional parameter configurations, yielding a desired behavior, increases the likelihood that gradient descent will converge to such parameters during training.

**Example 1** (Standard nonlinear readout maps). *The dominant choice for readout map $\phi(\cdot)$ is the exponential function $\exp(\cdot)$ (as in softmax attention), which is a positive monotonic non-decreasing function that is close to zero on the negative domain, i.e., $\phi(x) \to 0$ for $x < 0$. In this case, $\alpha_{i,j} \approx 0$ can be achieved by $q_i^\top h_{i,j} < 0$, which happens if the angle between the two vectors $q_i, h_{i,j}$ lies in $[\frac{\pi}{2} + n\pi, \frac{3\pi}{2} + n\pi]$ for any $n \in \mathbb{Z}^+$. This results in a large set $\mathcal{T}$, since many combinations of $q_i, h_{i,j}$ will lead to coefficients being approximately zero.*

Together, Lemmas 1 and 2 show the tradeoff in the choice of $\phi(\cdot)$ and offer a new perspective on the tradeoff between SSMs and Transformers (Gu, 2025): on the one hand, a nonlinear choice facilitates input selectivity; on the other, a linear choice facilitates efficient computation. To address this trade-off, various kernel approximation methods, i.e. $\phi \approx \psi_q(\cdot)\psi_k(\cdot)$, have been proposed in the literature, e.g. (Katharopoulos et al., 2020; Tsai et al., 2019; Arora et al., 2024), which try to approximate the behavior of $\phi(\cdot) = \exp(\cdot)$ as closely as possible, while striving to fulfill Principle 2. However, regardless of how exact the approximation is, there is a fundamental geometric issue with these approaches, as outlined in the next principle.

**Principle 2.1.** *If the readout map is of the form $\phi(\cdot) = \psi_q(\cdot)\psi_k(\cdot)$ (e.g.,kernel approximation), its zero-level set is of measure zero. Hence, by Principle 2, input selectivity is fragile to learn.*

**Corollary 2.1.** *Consider the kernel approximation of $\phi(\cdot) : \mathbb{R}^n \to \mathbb{R}^q$ to be $\tilde{\phi}(q_i^\top h_{i,j}) = \psi_q(q_i)^\top \psi_h(h_{i,j})$, with $\psi_q : \mathbb{R}^n \to \mathbb{R}^q$ and $\psi_h : \mathbb{R}^n \to \mathbb{R}^q$, such that it approximates the readout map $\phi(\cdot) \approx \tilde{\phi}(\cdot)$ in equation 3. Then, the zero-level set of the approximation $\tilde{\phi}(\cdot)$ is the singleton $\mathcal{Z} = \{0\}$ and by Lemma 2 the set of linear transformations $\mathcal{T}$ is small.*

By Principle 2 and the results from Lemma 2, choosing kernel approximations as readout maps hinders the capacity for input selectivity. To overcome this limitation, the kernel approximations $\psi_q(\cdot), \psi_k(\cdot)$ need to be designed carefully. While, setting $\psi_q, \psi_h$ to positive functions, e.g. $\psi_q(x) = \psi_h(x) = \text{elu}(x) + 1$ (Katharopoulos et al., 2020), preserves the positivity of the coefficients, it eliminates much of the crucial geometric information stored in $q_i$ and $h_{i,j}$, since the angle between these vectors is not preserved. To alleviate this issue, some approximation proposals induce a bias towards near-zero coefficients by explicitly using orthogonal feature maps, e.g. FAVOR+ (Choromanski et al., 2021) or random feature maps (Peng et al., 2021).

We note that Principle 2 presents general considerations for driving coefficients $\alpha_{i,j}$ to zero. Yet, by equation 2, $i$ coefficients are simultaneously computed using a single query $q_i$ and $i$ states (or transformed keys) $h_{i,j}$ with $j = 1, \ldots, i$. In what follows, we illustrate that achieving near-zero coefficients $\alpha_{i,j}$ *simultaneously* is a more restrictive setup than the general setting of Principle 2.

**Principle 2.2.** *If the readout map is the identity $\phi(\cdot) = \mathrm{Id}(\cdot)$, the number of coefficients that can be simultaneously suppressed by a nonzero query vector, is limited to $n-1$. Beyond this limit, suppression necessarily collapses to the trivial (zero) query vector.*

**Corollary 2.2.** *Let $y_L \in \mathbb{R}^{d_v}$ be the solution to equation 2, where $\alpha_{L,j} : \mathbb{R}^n \to \mathbb{R}$; $q_L \mapsto \alpha_{L,j}(q_L)$ is defined in equation 3 with identity readout map and normalization factor, i.e., $\phi(\cdot) = \mathrm{Id}(\cdot)$, $\eta_L = 1$. Consider the set of linearly independent states $\mathcal{H} = \{h_{L,t} \mid c_1 h_{L,1} + \cdots + c_t h_{L,t} = 0 \implies c_1 = \cdots = c_t = 0\}$. Then, a nonzero $q_L$ that achieves $\alpha_{L,j} = 0$, $\forall h_{L,j} \in \mathcal{H}$ exists if and only if $\dim\big(\mathrm{span}\{h_{L,j} \in \mathcal{H}\}\big) < n$. In particular, given a nonzero $q_L$, the measure $|\mathcal{H}| \leq n-1$.*

Principle 2.2 is primarily a limiting factor for identity readout maps, since zero coefficients $\alpha_{i,j}$ are only achieved for $q_i^\top h_{i,j} = 0$, i.e., $q_i$ and $h_{i,j}$ are orthogonal. For nonlinear readout maps (see Example 1), many configurations of $q_i$ and $h_{i,j}$ achieve $\alpha_{i,j} = 0$.

**Can learnable parameters save us?** We established that identity readout maps $\phi(\cdot) = \mathrm{Id}(\cdot)$ can suppress coefficients – if $q_i$, $h_{i,j}$ are orthogonal – but results in a much smaller set $\mathcal{T}$ than for nonlinear $\phi(\cdot)$, which in principle makes the learning problem harder. However, since $T_{h,i}$ in equation 4 is a function of $A_{i,j} := \prod_{t=j+1}^{i} A_t$ and $b_j$, these can be carefully parametrized and leveraged to achieve $h_{i,j} = 0$, which also results in zero coefficients. This is the central design paradigm of many SSM proposals, e.g. Dao & Gu (2024); Yang et al. (2025a). We elaborate more on how these choices can compensate for a linear readout map in the next section. However, it is important to note that although the choices of $A_{i,j}$ and $b_j$ can be used to suppress certain tokens, these are typically designed for input selectivity on the keys. For example, choosing $A_t = 1 - \lambda_t$, $b_j = \lambda_j$ (as in GRU Cho et al. (2014)),[3] enables selection of the state or key (equation 3). While this is straightforward to understand in an SSM setting, where the state aggregates the keys over time, our framework directly links this gating behavior to the coefficients $\alpha_{i,j}$. Choosing $\lambda_j = 0$ for a key $k_j$ results in suppression of this key in all future time steps, i.e., $\alpha_{i,j} = 0, \forall i \geq j$. Additionally, choosing $\lambda_t = 1$ or equivalently $A_t = 0$ for any $t \in [j, i]$, results in a zero state $h_{i,j}$ for all $i \geq t$, thus suppressing the coefficients. This shows that input selectivity design in SSMs, i.e., design of $A_t$, $b_j$, is a special case of the more general input selectivity design for coefficients considered in this work.

## 4.2 ON CHOOSING THE EVOLUTION MATRICES $A_t$

It is well-known that softmax attention cannot distinguish between two identical inputs that appear at different (time) positions in the sequence (Yun et al., 2020). The reason for this is the evolution matrix ($A_t = \mathbb{I}$) used in softmax attention and is commonly mitigated via positional embeddings (e.g. Su et al. (2024)). However, careful design of the evolution matrix enables processing of positional information in the attention mechanism, as evidenced by SSMs (Dao & Gu, 2024).

**Principle 3.** *If $A_t = \mathbb{I}$, no positional information is contained in the coefficients $\alpha_{i,j}$. This means for a sequence model choosing $A_t$ to be the identity, positional embeddings are needed.*

**Definition 1** (Positional Information). *$\alpha_{i,j}$ in equation 4 are said to have positional information, if for two identical inputs $x_j = x_{\bar{j}} = \bar{x}$, the resulting coefficients are not identical, i.e., $\alpha_{i,j} \neq \alpha_{i,\bar{j}}$.*

**Lemma 3.** *The coefficients $\alpha_{i,j}$ in equation 4 have positional information if and only if $A_t \neq \mathbb{I}_n, \forall t$.*

By Lemma 3, positional information can be embedded in the coefficients by designing $A_t$ appropriately, further strengthening our intuition why linear RNN/SSM proposals (such as Mamba-2, DeltaNet, mLSTM) do not require positional embeddings to learn in-context recall tasks (Arora et al., 2023; Poli et al., 2024; Okpekpe & Orvieto, 2025) or language (Gu & Dao, 2023). However, despite positional embeddings not being strictly necessary, using positional embeddings for RNNs and SSMs ($A_t \neq \mathbb{I}$) can be beneficial in practice (Morita, 2024).

Beyond enabling processing of positional information, the choices of $A_t$ can selectively suppress tokens under some circumstances. Specifically, as discussed in Principle 2, when the readout map $\phi(\cdot)$ is a linear function, $\alpha_{i,j} = 0$ is only achieved when $q_i$ and $h_{i,j}$ are orthogonal, or if one of

---

[3] A similar inverse relationship between $A_t$ and $b_j$ is found in many SSM proposals, see e.g. Dao & Gu (2024); De et al. (2024); Schlag et al. (2021).

the vectors $q_i^\top, h_{i,j}$ (or both) is zero. Since relying on the orthogonality of $q_i^\top$ and $h_{i,j}$ leads to harder learning problem (per Principle 2), many architectures that use a linear readout map for computational reasons (as per Principle 1), rely on dynamics equation 3 to achieve selectivity, e.g. Mamba-2. In these models, the right choice of the dynamics components $A_t$ and $b_j$ is crucial to ensure that $h_{i,j}$ can be effectively driven to zero. Additionally, the evolution matrices $A_t$ can be used to rotate non-orthogonal keys over time, to achieve $q_i^\top h_{i,j} = 0$, despite $q_i^\top k_j \neq 0$.

> **Principle 4.** *Assuming a linear readout map $\phi(\cdot)$, the structure imposed on the evolution matrices $A_t$ limits the operations (e.g. scaling, rotation) that can be performed on the keys.*

**Lemma 4.** *Imposing any of the following structures on $A_t$ in equation 3, results in the corresponding allowed transformations for the keys $\{k_j\}_{j=1}^i$:*

   a) $A_t = \lambda_t \mathbb{I}_n, \lambda_t \in \mathbb{R}, |\lambda_t| \leq 1$; allows scaling of keys (including flipping, if $\lambda_t < 0$) uniformly along all dimensions $n$,

   b) $A_t = diag(\lambda_t), \lambda_t \in \mathbb{R}^n, |\lambda_t^{(r)}| \leq 1$ where $\lambda^{(r)}$ denotes the $r$-th entry of $\lambda_t$; allows scaling of keys (including flipping, if $\lambda_t^{(r)} < 0$) separately along dimensions $n$,

   c) $A_t = \mathbb{I}_n - \beta_t \lambda_t \lambda_t^\top, \beta_t \in [0, 2], \lambda_t \in \mathbb{R}^n$, (Householder matrix); allows scaling and specific rotations of keys.

Lemma 4 provides guidance on how to design the evolution matrices $A_t$, depending on the transformations that should be enabled. Note that it is possible to combine these structures, e.g., combine a) and c) in Lemma 4 to obtain a scaled Householder matrix (Yang et al., 2025a). This list is not exhaustive and more general structures can be imposed on $A_t$ to achieve other transformations. Yet, these structures might be prohibited by current CUDA kernel implementations (Siems et al., 2025).

## 4.3   ON CHOOSING THE SCALING PARAMETERS $b_j$

As discussed in Section 4.1, the scaling parameters $b_j$ are often linked to the evolution matrices $A_t$ (see Table 1). While the need for scaling/normalization of keys has been established in previous architectural proposals, there is a lack of consensus on how to design $b_j$. The following result is based on classical signal propagation analyses (Glorot & Bengio, 2010; He et al., 2015).

> **Principle 5.** *Choosing the scaling parameter $b_j = \mathcal{O}(1/\sqrt{n})$ ensures that the dot product $q_i^\top h_{i,j}$ has $\mathcal{O}(1)$ variance; anything larger than this scale lets the variance grow with $n$.*

**Lemma 5.** *Consider two i.i.d. normally distributed random vectors $x_i, x_j \sim \mathcal{N}(0, \Sigma)$. Then, the dot product $q_i^\top h_{i,j}$ with $q_i, h_{i,j}$ defined in equation 4, has zero-mean and variance $Var(q_i^\top h_{i,j}) = tr(\Sigma_q \Sigma_h)$ with $\Sigma_q = T_q \Sigma T_q^\top$, $\Sigma_h = T_{h,i} \Sigma T_{h,i}^\top$. Assuming $\Sigma = \sigma \mathbb{I}_d$, both $\Sigma_q$ and $\Sigma_h$ are positive semi-definite and the variance of the dot product scales as $Var(q_i^\top h_{i,j}) = \mathcal{O}(n)$.*

Note that in Vaswani et al. (2017, Section 3.2.1), the statement of Lemma 5 is discussed informally. While the scaling factor choice $b_j = 1/\sqrt{n}$ is widely used for attention-based models, in SSMs or RNNs, $b_j$ is more often designed in combination with evolution matrices $A_t$. Since in these models, $A_t$ and $b_j$ have an inverse relationship and $A_t$ is typically designed to have eigenvalues close to the unit circle, this forces $b_j$ to be small. If the entangling of $A_t$ and $b_j$ is abandoned, Principle 5 provides a minimum scaling level that is essential for stable training of a sequence model.

**Example 2.** *Mamba-2 parameterizes both $A_t$ and $b_j$ with $\Delta_j$ (see Table 1), which is biased to lie in the range $\Delta_j \in [0.001, 0.1]$. This range can be thought of as $1/\sqrt{n}$ for $n \in [1e2, 1e6]$, thus fulfilling Principle 5 for a wide range of dimensions $n$.*

## 4.4   ON CHOOSING THE NORMALIZATION FACTORS $\eta_i$

While the scaling factors $b_j$ scale the keys, the normalization factors $\eta_i$ directly scale the $\alpha_{i,j}$s. Therefore, the normalization factors offer a way to re-scale all coefficients at time index $i$ ( amplifying large, reducing small) or normalize the linear combination at every time index individually.

> **Principle 6.** *If the readout map $\phi(\cdot)$ is unbounded and/or $A_t$ are not stable, the normalization factors $\eta_i$ need to be designed to counteract the growth in the coefficients.*

**Lemma 6.** *Consider a fixed index $j$ and let $s_i := q_i^\top h_{i,j}$, $\alpha_i = \phi(s_i)$ in equation 3. Let $\phi : \mathbb{R} \to [0, \infty)$ be nondecreasing and unbounded, and let $g : [0, \infty) \to (0, \infty)$ be an increasing and unbounded comparison function, such that $L := \limsup_{i \to \infty} \frac{\phi(s_i)}{g(s_i)} < \infty$. If $\eta_i$ is chosen such that $\liminf_{i \to \infty} \frac{\eta_i}{g(s_i)} := m > 0$, then the normalized coefficients are bounded: $\limsup_{i \to \infty} \frac{\alpha_i}{\eta_i} \leq \frac{L}{m}$.*

As an example, consider the mLSTM architecture (Beck et al., 2024), which allows unstable evolution matrices $A_t$ with eigenvalues outside the unit circle and an identity readout map. To fulfill Principle 6, the normalization factors $\eta_i$ need to be designed such that they grow linearly (or superlinearly) with the state $h_{i,j}$. The model does this by relying on a separate normalization state that uses the same dynamics as the state (see Table 1). An alternative option to design the normalization factors, is to choose $\eta_i$ as a function of the coefficients $\alpha_{i,j}$, i.e., $\eta_i = f(\alpha_{i,j})$. In this case, only linear (or superlinear) growth of $f$ is required to fulfill Principle 6 (see Remark 5). A simple choice of a superlinear $f$ is used in softmax attention, and is discussed in the following corollary.

**Corollary 6.1.** *Choosing $\eta_i = \sum_{j=1}^{i} \alpha_{i,j}$ in equation 2 constrains the normalized coefficients to $\tilde{\alpha}_{i,j} = \alpha_{i,j}/\eta_i \in [0, 1]$ and imposes $\sum_{j=1}^{i} \tilde{\alpha}_{i,j} = 1$.*

Corollary 6.1 additionally imposes a constraint on the linear combination in equation 2, which restricts the outputs $y_i$ to lie in a well-defined geometric space spanned by the value vectors. This fact and its implications are further discussed in Appendix A.

## 5 EXPERIMENTAL VALIDATION

In this section, we empirically validate the principles discussed in Section 4 on selected tasks of the MAD benchmark (Poli et al., 2024). Additional experiments are reported in Appendix D and all details of the experimental setup are discussed in Appendix E.

**Principle 1** Given that the principle concerns the implementation of sequence models, we show its implication on the model's throughput in Appendix D.

**Principle 2** Figure 2 shows the accuracy of different readout maps $\phi(\cdot)$ on the fuzzy in-context recall and selective copying tasks of MAD. The other parameters are fixed to $A_t = \mathbb{I}$, $b_j = 1/\sqrt{n}$, $\eta_i = \sum_j \alpha_{i,j}$, across all readout maps. Since input selectivity is crucial for both tasks, we plot accuracy against the fraction of coefficients close to 0 (near-zero set with $\epsilon = 0.001$) as well as the theoretical near-zero set of each $\phi(\cdot)$. *The results show, that more coefficients are set close to zero as the near-zero set of the readout map grows. For both tasks this increases performance.* Additional results and the results for Principles 2.1 & 2.2 are provided in Appendix D.

**Principle 3** We show that $A_t \neq \mathbb{I}_n$ can replace positional embeddings (PE) for the simplest choice of $A_t$. Figure 1 shows the accuracy of four models on the noisy in-context recall of MAD, where we set $b_j = 1/\sqrt{n}$, $\eta_i = \sum_j \alpha_{i,j}$, and vary $A_t \in \{\mathbb{I}_n, 0.95\,\mathbb{I}_n\}$, $\phi(\cdot) = \{\exp(\cdot), \text{softplus}(\cdot), \text{ReLU}(\cdot)\}$, and wether the models have PE or not. We choose a constant $A_t = 0.95\,\mathbb{I}_n$, since it directly connects to ALiBi (Press et al., 2022). While ALiBi enables positional information via additive biases to the attention matrix, $A_t = 0.95\,\mathbb{I}_n$ does the same but via multiplication. *The experiment shows that noisy in-context recall is solved by both PE and $A_t \neq \mathbb{I}_n$ for all readout maps. For $A_t = \mathbb{I}_n$*

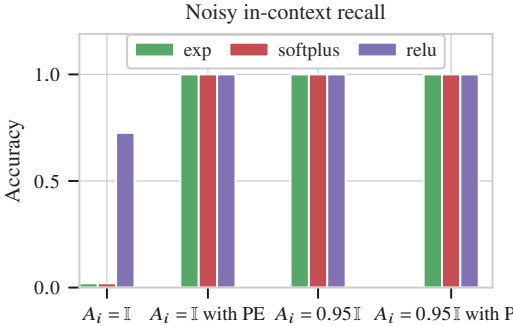

Figure 1: **(Principle 3)** Performance of two $A_t$ choices with and without positional embeddings (PE) on the noisy in-context recall task.

*without PE, only $\phi(\cdot) = ReLU(\cdot)$ achieves non-random performance but does not perform perfectly.*

**Principle 4** To show the effect of allowed transformations in $A_t$ on the performance, we ablate four structures imposed on $A_t$ – scalar, diagonal, Householder with keys $k_t$, and Householder with a learned vector $z_t$ – on the fuzzy in-context recall and selective copying tasks of MAD; the other parameters are fixed to $\phi(\cdot) = \text{Id}(\cdot)$, $\eta_i = 1$, $b_j = 1/\sqrt{n}$ across all choices of $A_t$. For each $A_t$, we additionally vary how the scalar and diagonal are parameterized (either using GLA (Yang et al., 2023)

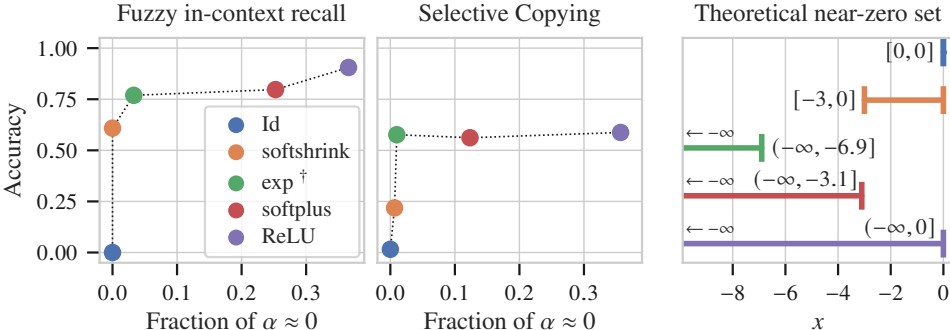

Figure 2: **(Principle 2)** Performance of different readout maps $\phi(\cdot)$ on two MAD tasks against the fraction of coefficients with near zero values ($|\alpha| \leq 0.001$; left & middle) and the theoretical near-zero sets of each readout map ($|\phi(x)| \leq 0.001$; right). The other parameters $A_t$, $b_j$, $\eta_i$ are fixed, thus the setting for $^\dagger$ is equivalent to softmax attention.

or Mamba-2 (Gu & Dao, 2023) parameterizations), and the scaling factor of the rotation vector in the Householder matrix (either $\beta_t = 2$ or learned from the input). *On both tasks, the performance generally increases as more transformations are allowed in $A_t$ (blue line). However, the parameterization of the involved parameters is important and good parameterizations can considerably improve performance (orange line).*

**Principles 5 & 6**  These principles are mainly concerned with training stability, which we show by ablating various choices of $b_j$ and $\eta_i$ in Appendix D.

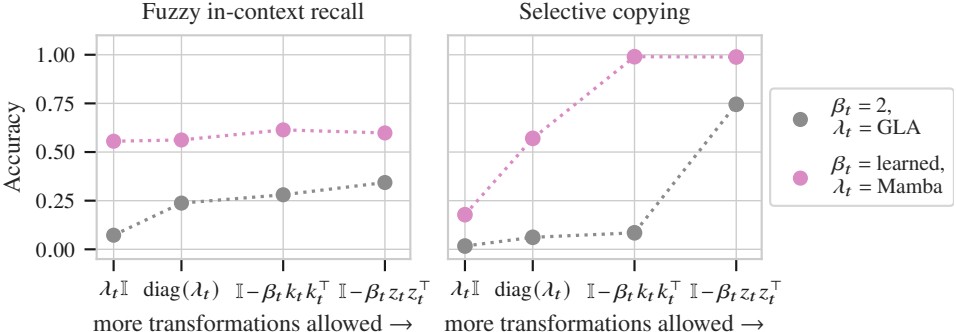

Figure 3: **(Principle 4)** Performance of four $A_t$ choices (scalar, diagonal, Householder with $k_t$, Householder with learned $z_t$) on two MAD tasks, with all other parameters fixed. The scalar/diagonal parameter(s) $\lambda_t$ are using either the GLA (gray) or Mamba-2 (magenta) parameterization and the Householder scaling $\beta_t$ is either fixed (gray) or learned (magenta).

## 6 CONCLUSIONS

This paper studies sequence models with a focus on the *coefficient dynamics* that multiply value vectors for a single layer. We view the sequence model outputs as linear combinations of past values with coefficients produced by autonomous linear dynamics with impulse inputs given by the keys. This view captures a broad class of existing sequence models, which can be recovered as a special case of this formalization. Furthermore, we show how studying the *coefficient dynamics* sheds light onto how normalization, geometric operations, and state updates influence input selectivity, efficient implementations, and training stability. These insights, formalized as mathematical results, unlock six design principles for sequence models that enable the tailored design of models. Experimental results validate these principles.

**Limitations:**  The present study is limited to single-layer setups and more work is needed to translate the proposed principles to multi-layer models. It also does not cover optimization considerations and the role of other components in the architecture, such as convolutions, gates, and specific positional embeddings, which could inform new principles or strengthen existing ones. Finally, the principles are validated on synthetic datasets and require additional experiments on real-world applications, such as language modeling.

## REPRODUCIBILITY STATEMENT

Several measures have been taken to ensure the reproducibility of our results. For theoretical results: all derivations for existing sequence models in Table 1 are provided in Appendix B, and proofs of all lemmas and corollaries stated in the paper are provided in Appendix C. For experimental results, all parameters and architectural choices are provided in Appendix E, for each experiment individually. We believe these measures allow full reproducibility of all results reported in this work.

## ACKNOWLEDGMENT OF AI-ASSISTED TOOLS

AI-assisted editing tools were used to improve grammar and style.

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

## CONTENTS OF APPENDIX

## A INTERPRETATION OF LINEAR COMBINATION CLASSES

Our perspective of viewing a sequence model's output as a linear combination of value vectors (equation 2) has important considerations. In particular, restrictions imposed on the normalized coefficients $\tilde{\alpha}_{i,j} = \frac{\alpha_{i,j}}{\eta_i}$ fully determine the output space of the model, as spanned by the value vectors. Specifically, any linear combination of the form equation 2 can be divided into four different classes depending on the restrictions imposed. These classes together with their restrictions, output space, and example architectures are provided in Table 2. The output space of these classes are also visualized in Figure 4.

Table 2: Summary of linear combination classes depending on the choice of coefficients.

| Type | Restrictions | Output Space | Example |
|---|---|---|---|
| Convex comb. | $\tilde{\alpha}_{i,j} \geq 0$ & $\sum_j \tilde{\alpha}_{i,j} = 1$ | Simplex | Softmax Attention |
| Conical comb. | $\tilde{\alpha}_{i,j} \geq 0$ | Positive Orthant | Normalized Attention |
| Affine comb. | $\sum_j \tilde{\alpha}_{i,j} = 1$ | Hyperplane | GPAM (Heo & Choi, 2024) |
| Linear comb. | None | $\mathbb{R}^{d_v}$ | Mamba-2, DeltaNet |

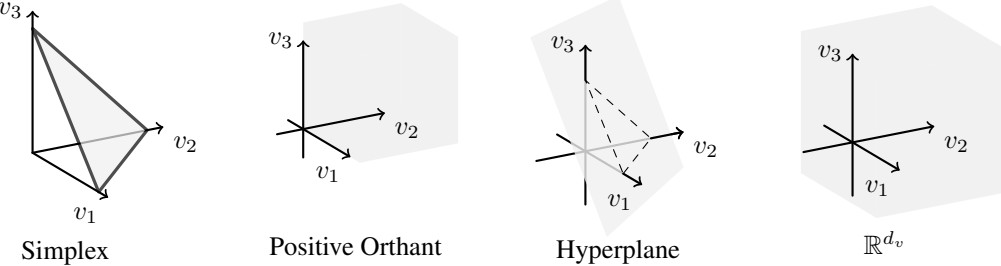

Figure 4: Visual representation of output spaces referenced in Table 2.

The structure of the output space defined by the normalized coefficients $\tilde{\alpha}_{i,j}$ and the value vectors $\{v_1, \ldots, v_i\}$ can be linked to a model's performance, however this area remains underexplored. For instance, the output of softmax attention lies in the convex hull (simplex) of previous value vectors, which has been linked to various performance traits, such as in-context learning (von Oswald et al., 2023),[4] and failure to represent basic logical operators (XOR) (Richter & Wattenhofer, 2020). More research into the role of linear combination classes and their effect on a model's performance or properties is needed to better understand the implications, but we believe this paper has shown the advantages of viewing sequence models through the lens of linear combinations.

## B LINEAR COMBINATION COEFFICIENTS IN EXISTING ARCHITECTURES

In the following, we derive the coefficient dynamics of all architectures listed in Table 1. To do this, we make use of a model's representation for a single output token (e.g. equation 5) and equation 4 to transfer each model into a linear combination (equation 2).

---

[4]Their derivation of the in-context gradient step hinges on the probability simplex imposed by the softmax operation.

## B.1 SOFTMAX ATTENTION

**Lemma 7** (Coefficient Dynamics of Softmax Attention). *Softmax attention (Vaswani et al., 2017, eq. 1)*

$$y_i = \sum_{j=0}^{i} \frac{\exp(\frac{q_i^\top k_j}{\sqrt{n}}) v_j}{\sum_{j=1}^{i} \exp(\frac{q_i^\top k_j}{\sqrt{n}})} \tag{5}$$

*can be written in the form of equation 2, by setting $A_i = \mathbb{I}_n$, $b_j = \frac{1}{\sqrt{n}}$, $\phi(\cdot) = \exp(\cdot)$, and $\eta_i = \sum_{j=1}^{i} \exp(\frac{q_i^\top k_j}{\sqrt{n}})$. The resulting normalized coefficients satisfy $\tilde{\alpha}_{i,j} \geq 0$ and $\sum_{j=1}^{i} \tilde{\alpha}_{i,j} = 1$, placing the output in the convex hull of the value vectors.*

*Proof.* First, identify the following parameters, by comparing equation 5 to equation 2:

$$\alpha_{i,j} = \exp(\frac{q_i^\top k_j}{\sqrt{n}}), \quad \eta_i = \sum_{j=1}^{i} \exp(\frac{q_i^\top k_j}{\sqrt{n}}).$$

Then, the parameter choices $A_i = \mathbb{I}_n$, $b_j = \frac{1}{\sqrt{n}}$, and $\phi(\cdot) = \exp(\cdot)$ ensure the coefficients $\alpha_{i,j}$ align with equation 4.

The properties $\tilde{\alpha}_{i,j} \geq 0$ follow from $\exp(\cdot) \geq 0$, and $\sum_{j=1}^{i} \tilde{\alpha}_{i,j} = 1$ follows by construction of the normalization factor $\eta_i$. Therefore, the output $y_i = \sum_{j=1}^{i} \tilde{\alpha}_{i,j} v_j$ lies in the convex hull of $\{v_1, \ldots, v_i\}$. $\square$

## B.2 LINEAR ATTENTION

**Lemma 8** (Coefficient Dynamics of Linear Attention). *Linear attention (Katharopoulos et al., 2020, eq. 4)*

$$y_i = \sum_{j=1}^{i} \frac{\psi(q_i)^\top \psi(k_j/\sqrt{n}) v_j}{\sum_{j=0}^{i} \psi(q_i)^\top \psi(k_j/\sqrt{n})} \tag{6}$$

*with $\psi(\cdot) \geq 0$, can be written in the form of equation 2 by setting $A_i = \mathbb{I}_n$, $b_j = \frac{1}{\sqrt{n}}$, $\phi(\cdot) = \psi(\cdot)\psi(\cdot)$, and $\eta_i = \sum_{j=1}^{i} \psi(q_i)^\top \psi(k_j/\sqrt{n})$. The resulting normalized coefficients satisfy $\tilde{\alpha}_{i,j} \geq 0$ and $\sum_{j=1}^{i} \tilde{\alpha}_{i,j} = 1$, placing the output in the convex hull of the value vectors.*

*Proof.* First, identify the following parameters, by comparing equation 6 to equation 2:

$$\alpha_{i,j} = \psi(q_i)^\top \psi(\frac{k_j}{\sqrt{n}}), \quad \eta_i = \sum_{j=1}^{i} \psi(q_i)^\top \psi(k_j/\sqrt{n}).$$

Then, the parameter choices $A_i = \mathbb{I}_n$, $b_j = \frac{1}{\sqrt{n}}$, and $\phi(\cdot) = \psi(\cdot)\psi(\cdot)$[5] ensure the coefficients $\alpha_{i,j}$ align with equation 4. Note that $\psi(\cdot)$ can be seen as preprocessing steps of the queries and keys, therefore allowing to write the coefficients as $\alpha_{i,j} = \tilde{q}_i^\top \tilde{k}_j/\sqrt{n}$ with $\tilde{q}_i = \psi(q_i)$ and $\tilde{k}_j = \psi(k_j)$, which reveals that the readout map is effectively the identity as discussed in Section 4.1.

The properties $\tilde{\alpha}_{i,j} \geq 0$ follow from $\psi(\cdot) \geq 0$, and $\sum_{j=1}^{i} \tilde{\alpha}_{i,j} = 1$ follows by construction of the normalization factor $\eta_i$. Therefore, the output $y_i = \sum_{j=1}^{i} \tilde{\alpha}_{i,j} v_j$ lies in the convex hull of $\{v_1, \ldots, v_i\}$. $\square$

---

[5]Katharopoulos et al. (2020) uses $\psi(x) = \text{elu}(x) + 1$, but there exist many more proposals for $\psi(\cdot)$ in the literature.

### B.3 NORMALIZED ATTENTION

**Lemma 9** (Coefficient Dynamics of Normalized Attention). *Normalized attention (Sieber et al., 2024, eq. 22)*

$$y_i = \sum_{j=1}^{i} \frac{\frac{q_i^\top k_j}{\sqrt{n}} v_j}{\exp(W_\eta x_i)} \tag{7}$$

*can be written in the form of equation 2 by setting $A_i = \mathbb{I}_n$, $b_j = \frac{1}{\sqrt{n}}$, $\phi(\cdot) = Id(\cdot)$, and $\eta_i = \exp(W_\eta x_i)$. The resulting normalized coefficients impose no additional constraints on the output.*

*Proof.* First, identify the following parameters, by comparing equation 7 to equation 2:

$$\alpha_{i,j} = \frac{q_i^\top k_j}{\sqrt{n}}, \quad \eta_i = \exp(W_\eta x_i).$$

Then, the parameter choices $A_i = \mathbb{I}_n$, $b_j = \frac{1}{\sqrt{n}}$, and $\phi(\cdot) = Id(\cdot)$ ensure the coefficients $\alpha_{i,j}$ align with equation 4.

Note that positive preprocessing functions as in linear attention ($\psi(\cdot) \geq 0$) are allowed, which would constrain the normalized coefficients to $\tilde{\alpha}_{i,j} \geq 0$. Otherwise, there are no restrictions on the resulting normalized coefficients $\tilde{\alpha}_{i,j}$, placing the output in the full space $\mathbb{R}^{d_v}$. $\square$

### B.4 GATED LINEAR ATTENTION (GLA)

**Lemma 10** (Coefficient Dynamics of Gated Linear Attention (GLA)). *Gated linear attention (Yang et al., 2023, eq. 3)[6]*

$$s_t = diag(\alpha_t)s_{t-1} + k_t v_t \tag{8}$$

*can be written in the form of equation 2 by setting $A_i = diag(\alpha_i)$, $b_j = \frac{1}{\sqrt{n}}$, $\phi(\cdot) = Id(\cdot)$, and $\eta_i = 1$. The resulting normalized coefficients impose no additional constraints on the output.*

*Proof.* First, we need to unroll equation 8 to get an elementwise convolutional representation (see Sieber et al. (2024); Dao & Gu (2024)):

$$s_i = \sum_{j=1}^{i} \left( \prod_{t=j+1}^{i} diag(\alpha_t) \right) k_j v_j,$$

before computing the output by multiplying the queries:

$$o_i = q_i^\top \sum_{j=1}^{i} \left( \prod_{t=j+1}^{i} diag(\alpha_t) \right) k_j v_j.$$

Note that Yang et al. (2023) implicitly assumes that the keys are scaled by $\frac{1}{\sqrt{n}}$,[7] and rearranging above equation yields

$$y_i = \sum_{j=1}^{i} q_i^\top \left( \prod_{t=j+1}^{i} diag(\alpha_t) \right) \frac{k_j}{\sqrt{n}} v_j.$$

We identify the following parameters by comparing to equation 2

$$\alpha_{i,j} = q_i^\top \left( \prod_{t=j+1}^{i} diag(\alpha_t) \right) \frac{k_j}{\sqrt{n}}, \quad \eta_i = 1.$$

---

[6]Here, we assume the vector product $k_t v_t$ is well-defined. This is explicit in Yang et al. (2023), but only needed if recurrences are computed, which is not the case in this paper. Therefore, assume here that $k_t v_t$ computes $k_t v_t^\top$ and that the product $q_i^\top s_i$ is transposed after summation. Then, this is the equivalent setting to (Yang et al., 2023, eq. 3) but with all quantities transposed.

[7]See codebase: https://github.com/fla-org/flash-linear-attention.

Then, the parameter choices $A_i = \text{diag}(\alpha_i)$,[8] $b_j = \frac{1}{\sqrt{n}}$, and $\phi(\cdot) = \text{Id}(\cdot)$ ensure the coefficients $\alpha_{i,j}$ align with equation 4. The resulting normalized coefficients $\tilde{\alpha}_{i,j}$ are unrestricted in sign and magnitude, placing the output in the full space $\mathbb{R}^{d_v}$. □

## B.5 MAMBA-2

**Lemma 11** (Coefficient Dynamics of Mamba-2). *Mamba-2 (Dao & Gu, 2024)*

$$y_i = \sum_{j=1}^{i} q_i^\top \left( \prod_{t=j+1}^{i} \exp(-\Delta_t A) \right) \Delta_j k_j v_j \tag{9}$$

*can be written in the form of equation 2 by setting $A_i = \exp(-\Delta_i A)$, $b_j = \Delta_j$, $\phi(\cdot) = Id(\cdot)$, and $\eta_i = 1$. The resulting normalized coefficients impose no additional constraints on the output.*

*Proof.* First, note that equation 9 is directly obtained from state-space duality (SSD; see Sieber et al. (2024); Dao & Gu (2024)). We identify the following parameters by comparing to equation 2

$$\alpha_{i,j} = q_i^\top \left( \prod_{t=j+1}^{i} \exp(-\Delta_t A) \right) \Delta_j k_j, \quad \eta_i = 1.$$

Then, the parameter choices $A_i = \exp(-\Delta_i A)$, $b_j = \Delta_j$, and $\phi(\cdot) = \text{Id}(\cdot)$ ensure the coefficients $\alpha_{i,j}$ align with equation 4.

Note that the *discretization step* $\Delta_j$ is computed from the input as $\Delta_j = \text{softplus}(W_\Delta x_i + \beta)$. The resulting normalized coefficients $\tilde{\alpha}_{i,j}$ are unrestricted in sign and magnitude, placing the output in the full space $\mathbb{R}^{d_v}$. □

## B.6 DELTANET

**Lemma 12** (Coefficient Dynamics of DeltaNet). *DeltaNet (Schlag et al., 2021, eq. 24)[9]*

$$s_t = s_{t-1} + \beta_t k_t (v_t - k_t^\top s_{t-1}) = (\mathbb{I} - \beta_t k_t k_t^\top) s_{t-1} + \beta_t k_t v_t, \tag{10}$$

*can be written in the form of equation 2 by setting $A_i = \mathbb{I} - \beta_i k_i k_i^\top$, $b_j = \frac{\beta_j}{\sqrt{n}}$, $\phi(\cdot) = Id(\cdot)$, and $\eta_i = 1$. The resulting normalized coefficients impose no additional constraints on the output.*

*Proof.* First, we need to unroll equation 10 to get an elementwise convolutional representation (see Sieber et al. (2024); Dao & Gu (2024)):

$$s_i = \sum_{j=1}^{i} \left( \prod_{t=j+1}^{i} \mathbb{I} - \beta_t k_t k_t^\top \right) \beta_j k_j v_j,$$

before computing the output by multiplying the queries:

$$o_i = q_i^\top \sum_{j=1}^{i} \left( \prod_{t=j+1}^{i} \mathbb{I} - \beta_t k_t k_t^\top \right) \beta_j k_j v_j.$$

Note that Schlag et al. (2021, Section 3.1) assumes that the keys are scaled by $\frac{1}{\sqrt{n}}$, and rearranging above equation yields

$$y_i = \sum_{j=1}^{i} q_i^\top \left( \prod_{t=j+1}^{i} \mathbb{I} - \beta_t k_t k_t^\top \right) \frac{\beta_j}{\sqrt{n}} k_j v_j.$$

---

[8]In Yang et al. (2023) the evolution matrix is computed as the low-rank parameterization $\alpha_i = \sigma(W_\alpha^2 W_\alpha^1 x_i + b_\alpha)^{\frac{1}{\tau}}$ with $\tau = 16$.

[9]Here, we assume the vector product $k_t v_t$ is well-defined. This is explicit in Schlag et al. (2021), but only needed if recurrences are computed, which is not the case in this paper. Therefore, assume here that $k_t v_t$ computes $k_t v_t^\top$ and that the product $q_i^\top s_i$ is transposed after summation. Then, this is the equivalent setting to (Schlag et al., 2021, eq. 24) but with all quantities transposed.

We identify the following parameters by comparing to equation 2

$$\alpha_{i,j} = q_i^\top \left( \prod_{t=j+1}^{i} \mathbb{I} - \beta_t k_t k_t^\top \right) \frac{\beta_j}{\sqrt{n}} k_j, \quad \eta_i = 1.$$

Then, the parameter choices $A_i = \mathbb{I} - \beta_i k_i k_i^\top$, $b_j = \frac{\beta_j}{\sqrt{n}}$,[10] and $\phi(\cdot) = \text{Id}(\cdot)$ ensure the coefficients $\alpha_{i,j}$ align with equation 4.

As discussed in Schlag et al. (2021, Section 4.2), the keys $k_i$ need to be normalized (i.e. $\|k_i\| = 1$), otherwise the evolution matrix $A_i$ is not a proper Householder matrix and its eigenvalues can lie outside the unit circle leading to instability. This normalization can be seen as a preprocessing step of the keys. Additionally, DeltaNet allows preprocessing of keys and queries with positive functions (similar to linear attention, i.e., $\psi(\cdot) \geq 0$), which would constrain the normalized coefficients to $\tilde{\alpha}_{i,j} \geq 0$. Otherwise, there are no restrictions on the resulting normalized coefficients $\tilde{\alpha}_{i,j}$, placing the output in the full space $\mathbb{R}^{d_v}$. □

### B.7 GATED DELTANET

**Lemma 13** (Coefficient Dynamics of Gated DeltaNet). *Gated DeltaNet (Yang et al., 2025a, eq. 10)*[11]

$$s_t = \alpha_t(\mathbb{I} - \beta_t k_t k_t^\top)s_{t-1} + \beta_t k_t v_t, \tag{11}$$

*can be written in the form of equation 2 by setting $A_i = \alpha_t(\mathbb{I} - \beta_i k_i k_i^\top)$, $b_j = \frac{\beta_j}{\sqrt{n}}$, $\phi(\cdot) = Id(\cdot)$, and $\eta_i = 1$. The resulting normalized coefficients impose no additional constraints on the output.*

*Proof.* First, we need to unroll equation 11 to get an elementwise convolutional representation (see Sieber et al. (2024); Dao & Gu (2024)):

$$s_i = \sum_{j=1}^{i} \left( \prod_{t=j+1}^{i} \alpha_t(\mathbb{I} - \beta_t k_t k_t^\top) \right) \beta_j k_j v_j,$$

before computing the output by multiplying the queries:

$$o_i = q_i^\top \sum_{j=1}^{i} \left( \prod_{t=j+1}^{i} \alpha_t(\mathbb{I} - \beta_t k_t k_t^\top) \right) \beta_j k_j v_j.$$

Note that Yang et al. (2025a) implicitly assumes that the keys are scaled by $\frac{1}{\sqrt{n}}$,[12] and rearranging above equation yields

$$y_i = \sum_{j=1}^{i} q_i^\top \left( \prod_{t=j+1}^{i} \alpha_t(\mathbb{I} - \beta_t k_t k_t^\top) \right) \frac{\beta_j}{\sqrt{n}} k_j v_j.$$

We identify the following parameters by comparing to equation 2

$$\alpha_{i,j} = q_i^\top \left( \prod_{t=j+1}^{i} \alpha_t(\mathbb{I} - \beta_t k_t k_t^\top) \right) \frac{\beta_j}{\sqrt{n}} k_j, \quad \eta_i = 1.$$

---

[10]The *writing strength* is computed as $\beta_i = \sigma(W_\beta)$, with $\sigma(\cdot)$ the sigmoid function. It can be multiplied by 2 to allow negative eigenvalues of $A_i$.

[11]Here, we assume the vector product $k_t v_t$ is well-defined. This is explicit in Yang et al. (2025a), but only needed if recurrences are computed, which is not the case in this paper. Therefore, assume here that $k_t v_t$ computes $k_t v_t^\top$ and that the product $q_i^\top s_i$ is transposed after summation. Then, this is the equivalent setting to (Yang et al., 2025a, eq. 10) but with all quantities transposed.

[12]See codebase: https://github.com/fla-org/flash-linear-attention.

Then, the parameter choices $A_i = \alpha_i(\mathbb{I} - \beta_i k_i k_i^\top)$,[13] $b_j = \frac{\beta_j}{\sqrt{n}}$,[14] and $\phi(\cdot) = \text{Id}(\cdot)$ ensure the coefficients $\alpha_{i,j}$ align with equation 4.

As discussed in DeltaNet, the keys $k_i$ need to be normalized (i.e. $\|k_i\| = 1$), otherwise the evolution matrix $A_i$ is not a proper Householder matrix and its eigenvalues can lie outside the unit circle leading to instability. This normalization can be seen as a preprocessing step of the keys. The resulting normalized coefficients $\tilde{\alpha}_{i,j}$ are unrestricted in sign and magnitude, placing the output in the full space $\mathbb{R}^{d_v}$. $\square$

### B.8 MLSTM

**Lemma 14** (Coefficient Dynamics of mLSTM). *mLSTM (Beck et al., 2024, eq. 19 - 27)*[15]

$$s_t = \exp(f_t)s_{t-1} + \exp(i_t)k_t v_t, \tag{12a}$$

$$y_t = \frac{o_t}{\eta_t} q_t^\top s_t \tag{12b}$$

*can be written in the form pf equation 2 by setting $A_i = \exp(f_t)$, $b_j = \frac{\exp(i_t)}{\sqrt{n}}$, $\phi(\cdot) = Id(\cdot)$, and $\eta_i = \frac{\eta_i}{o_i}$. The resulting normalized coefficients impose no additional constraints on the output.*

*Proof.* First, we need to unroll equation 12a to get an elementwise convolutional representation (see Sieber et al. (2024); Dao & Gu (2024)):

$$s_i = \sum_{j=1}^{i} \left( \prod_{t=j+1}^{i} \exp(f_t) \right) \exp(i_j) k_j v_j,$$

before computing the output as

$$y_i = \frac{o_i}{\eta_i} q_i^\top \sum_{j=1}^{i} \left( \prod_{t=j+1}^{i} \exp(f_t) \right) \exp(i_j) k_j v_j.$$

Note that Beck et al. (2024, eq. 23) assumes that the keys are scaled by $\frac{1}{\sqrt{n}}$, and rearranging above equation yields

$$y_i = \sum_{j=1}^{i} \frac{o_i}{\eta_i} q_i^\top \left( \prod_{t=j+1}^{i} \exp(f_t) \right) \frac{\exp(i_j)}{\sqrt{n}} k_j v_j.$$

We identify the following parameters by comparing to equation 2

$$\alpha_{i,j} = q_i^\top \left( \prod_{t=j+1}^{i} \exp(f_t) \right) \frac{\exp(i_j)}{\sqrt{n}} k_j, \quad \eta_i = \frac{\eta_i}{o_i}.$$

Then, the parameter choices $A_i = \exp(f_i)$, $b_j = \frac{\exp(i_j)}{\sqrt{n}}$, and $\phi(\cdot) = \text{Id}(\cdot)$ ensure the coefficients $\alpha_{i,j}$ align with equation 4.

Note that the forget and input gates $f_i$, $i_i$ are computed by linear input projections and that $\exp(f_i)$, $\exp(i_i)$ need to be regularized for numerical stability, since both can grow exponentially fast. In mLSTM this is achieved by multiplying $\exp(m_{i-1} - m_i)$ and $\exp(m_i)$ to the forget and input gates, respectively, where $m_i$ is computed via another recurrence (for all details see Beck et al. (2024)). The resulting normalized coefficients $\tilde{\alpha}_{i,j}$ are unrestricted in sign and magnitude, placing the output in the full space $\mathbb{R}^{d_v}$. $\square$

---

[13]$\alpha_i$ is computed equivalently to $\exp(-\Delta_i A)$ in Mamba-2.

[14]The *writing strength* is computed as $\beta_i = \sigma(W_\beta)$, with $\sigma(\cdot)$ the sigmoid function. It can be multiplied by 2 to allow negative eigenvalues of $A_i$.

[15]Here, we assume the vector product $k_t v_t$ is well-defined. This is explicit in Beck et al. (2024), but only needed if recurrences are computed, which is not the case in this paper. Therefore, assume here that $k_t v_t$ computes $k_t v_t^\top$ and that the product $q_i^\top s_i$ is transposed after summation. Then, this is the equivalent setting to (Beck et al., 2024, eq. 19) but with all quantities transposed.

# C  PROOFS

This appendix is split into two parts: Section C.1 provides the proofs of all lemmas in the main text and Section C.2 provides the proofs of all corollaries in the main text. For convenience, the lemmas and corollaries are copied before the proofs.

## C.1  LEMMAS

### C.1.1  LEMMA 1

**Lemma 1.** *A recurrent formulation of equation 3 with finite memory (state) in $\mathbb{R}^{n \times d_v}$, which allows simultaneous computation of $\alpha_{i,j}$, exists if and only if $\phi(\cdot) : \mathbb{R} \to \mathbb{R}$ is a linear map.*

*Proof.* We prove the two directions, i.e., sufficiency and necessity, separately.

**(Necessity.)** If $\phi(\cdot)$ is linear, i.e., $\phi(x) = ax$, $a \in \mathbb{R}$, then

$$y_i = \sum_{j=1}^{i} \frac{a\, q_i^\top h_{i,j}}{\eta_i}\, v_j = \frac{a\, q_i^\top}{\eta_i} \sum_{j=1}^{i} h_{i,j}\, v_j^\top,$$

where the second equality is due to linearity and equivalent to the reformulation of linear attention in Katharopoulos et al. (2020). Then, expanding $h_{i,j}$ using equation 3 yields

$$y_i = \frac{a\, q_i^\top}{\eta_i} \sum_{j=1}^{i} \left( \prod_{t=j+1}^{i} A_t \right) b_j k_j\, v_j^\top.$$

Define the $n \times d_v$ matrix $S_i := k_i v_i^\top$ and note that $S_i$ admits recurrent updates to compute the sum, i.e.,

$$S_i = A_i S_{i-1} + b_i k_i v_i^\top,$$

$$y_i = \frac{a\, q_i^\top}{\eta_i} S_i,$$

Therefore, a recurrent implementation with finite state (memory) $S_i \in \mathbb{R}^{n \times d_v}$ exists if $\phi(\cdot)$ is linear.

**(Sufficiency.)** Assume a finite state (memory) recurrent implementation exists, i.e.,

$$S_i = A_i S_{i-1} + b_i k_i v_i^\top, \tag{13a}$$

$$y_i = \frac{a\, q_i^\top}{\eta_i} S_i, \tag{13b}$$

where $a \in \mathbb{R}$ is a scaling factor. We prove that $\phi(\cdot)$ must be linear by contradiction, thus suppose $\phi(\cdot)$ is any nonlinear function on $\mathbb{R}$, such that the output $y_i$ can be computed as

$$y_i = \sum_{j=1}^{i} \frac{\phi(q_i^\top h_{i,j})}{\eta_i}\, v_j. \tag{14}$$

Hence, the expression from $y_i$ derived from rolling out equation 13 must be equal to equation 14:

$$\frac{a\, q_i^\top}{\eta_i} \sum_{j=1}^{i} \left( \prod_{t=j+1}^{i} A_t \right) b_j k_j\, v_j^\top = \sum_{j=1}^{i} \frac{phi(q_i^\top h_{i,j})}{\eta_i}\, v_j,$$

and by simplifying the equation above with the definition of $h_{i,j}$ (equation 3) we have

$$\sum_{j=1}^{i} \frac{a\, q_i^\top h_{i,j}}{\eta_i} v_j^\top = \sum_{j=1}^{i} \frac{\phi(q_i^\top h_{i,j})}{\eta_i}\, v_j.$$

This equality does not hold for any nonlinear function $\phi(\cdot)$, but only a linear one, thus concluding the proof.  □

### C.1.2 LEMMA 2

**Lemma 2.** *Let $\phi : \mathbb{R} \to \mathbb{R}$ be the readout map in equation 3 with connected zero-level set $\mathcal{Z} = \{z \in \mathbb{R} \mid \phi(z) = 0\}$, and let $z = q_i^\top h_{i,j}$, with $T_q, T_{h,i}$ defined in equation 4. Let $\mathcal{T}$ be the set of linear transformations that achieve the zero-level set, i.e., $\mathcal{T} = \{(T_q, T_{h,i}) \mid q_i^\top h_{i,j} \in \mathcal{Z} \text{ s.t. } q_i = T_q x_i, h_{i,j} = T_{h,i} x_j\}$ for any pair of inputs $x_i, x_j \in \mathbb{R}^d$. Then, the measure of $\mathcal{T}$ is directly proportional to the measure of the zero-level set $\mathcal{Z}$, i.e., $|\mathcal{T}| = c|\mathcal{Z}|$ with $c > 0$.*

*Proof.* Given that the set $\mathcal{Z}$ is connected on $\mathbb{R}$, the set is defined by the interval $\mathcal{Z} = [\underline{z}, \overline{z}] \subset \mathbb{R}$. Then, since $T_q, T_{h,i} \in \mathbb{R}^{n \times d}$, we consider the Euclidean space $E^{2nd}$ of pairs $(T_q, T_{h,i})$, equipped with the Lebesgue measure $\lambda(E)$. We define the map

$$F : E \to \mathbb{R}, \qquad F(T_q, T_{h,i}) = (T_q x_i)^\top (T_{h,i} x_j),$$

which is smooth and its derivative $\nabla F(T_q, T_{h,i})$ is non-zero for $x_i \neq 0$ and $x_j \neq 0$; we treat the trivial case of either $x_i = 0$ or $x_j = 0$ at the end of the proof.

Then, for each $z \in [\underline{z}, \overline{z}]$, we construct the subset $\mathcal{T}_z$ as a fiber[16] of $F$, i.e.,

$$\mathcal{T}_z := F^{-1}(\{z\}) = \{(T_q, T_{h,i}) \in E \mid (T_q x_i)^\top (T_{h,i} x_j) = z\}.$$

Therefore, the set of all linear transformations that achieve a value in $\mathcal{Z}$ is

$$\mathcal{T} = F^{-1}(\mathcal{Z}) = \bigcup_{z \in \mathcal{Z}} \mathcal{T}_z.$$

To prove the lemma, we need to compute the Lebesgue measure of $\mathcal{T}$[17], i.e., $|\mathcal{T}| = \int_E \mathbf{1}_{F^{-1}(\mathcal{Z})} \lambda(t)$, where $t = (T_q, T_{h,i})$ and $\mathbf{1}_{F^{-1}(\mathcal{Z})}$ is the indicator function selecting all $t \in \mathcal{T}$. Since $x_i, x_j \neq 0$, $F$ is smooth and $\nabla F \neq 0$ on $E$, we can apply the coarea formula (Federer, 1959), i.e.,

$$\int_E \mathbf{1}_{F^{-1}(\mathcal{Z})}(t)\, \lambda(t) = \int_\mathcal{Z} \left( \int_{F^{-1}(z)} \frac{1}{\|\nabla F(t)\|} \, dH^{2nd-1}(t) \right) dz$$

where $H^{2nd-1}$ is the $(2nd-1)$-dimensional Hausdorff measure and $\|\nabla F(t)\|$ is the Euclidean norm of the gradient of $F$ at $t$. We then define

$$g(z) := \int_{F^{-1}(z)} \frac{1}{|\nabla F(t)|} \, d\mathcal{H}^{2nd-1}(t),$$

which denotes the size of each subset (fiber) $\mathcal{T}_z$. Therefore, the measure of $\mathcal{T}$ is

$$|\mathcal{T}| = \int_\mathcal{Z} g(z) dz,$$

and by choosing the constant $c$ as

$$c = \frac{1}{|\mathcal{Z}|} \int_\mathcal{Z} g(z) dz,$$

gives the desired relation $|\mathcal{T}| = c|\mathcal{Z}|$.

In case $x_i, x_j = 0$, the dot product $q_i^\top h_{i,j}$ is trivially zero, thus $F(t) = 0, \forall t = (T_q, T_{h,i}) \in E$. In this case, $|\mathcal{T}| = 0$, if $0 \notin \mathcal{Z}$, i.e., empty because $z = 0$ is not contained in the zero-level set, or $|\mathcal{T}| = |E|$, i.e., the measure of the Euclidean space $E$. $\square$

---

[16]The fiber $f^{-1}(y)$ of a function $f$ is the preimage of the singleton $y$.

[17]Intuitively, the volume of the set $\mathcal{T}$.

**Remark 2.** *To intuitively understand Lemma 2 and its proof, consider the simplest case: $q_i = T_q x_i = a x_i \in \mathbb{R}$, $h_{i,j} = T_{h,i} x_j = b x_j \in \mathbb{R}$, thus all variables are scalars. Additionally, consider a readout function $\phi(\cdot)$ with zero-level set $\mathcal{Z} = [-1, 0]$. In this setting, we want to find all pairs $t = (a, b)$ such that $ab(x_i x_j) = z$ and we define $X = x_i x_j \neq 0$. Then, all $t$ that achieve $z$ in the zero-level set, need to lie on the hyperbola*

$$a \cdot b = \frac{z}{X}.$$

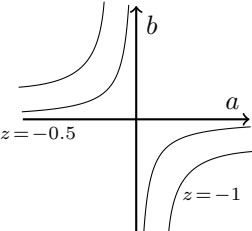

*This hyperbola exactly describes the set $\mathcal{T}_z$ in the proof and there are infinitely many combinations $(a, b)$ that lie in this set. To get to $\mathcal{T}$, we need to integrate over the zero-level set $\mathcal{Z}$ to account for all possible $z$. These sets (hyperbolas) are also visualized in the figure on the right.*

### C.1.3 LEMMA 3

**Lemma 3.** *The coefficients $\alpha_{i,j}$ in equation 4 have positional information if and only if $A_i \neq \mathbb{I}_n, \forall i$.*

*Proof.* Consider the definition of the coefficients (equation 4):

$$\alpha_{i,j} = \phi \left( q_i^\top \left( \prod_{t=j+1}^{i} A_t \right) b_j k_j \right).$$

We first consider the case $A_t = \mathbb{I}_n, \forall t$, then

$$\alpha_{i,j} = \phi \left( q_i^\top b_j k_j \right).$$

Assuming the scaling factors are computed from the input $b_j = f(x_j)$, the factors $b_j$ contain no positional information. Specifically, because $b_j$ is computed from the same input $x_j$ as $k_j$, we can rewrite the key using a change of variable $\tilde{k}_j = b_j k_j = g(x_j)$, i.e., the new key is computed from a single input $x_j$, albeit in a nonlinear fashion. Therefore, we consider the simplified coefficients

$$\bar{\alpha}_{i,j} = \phi \left( q_i^\top \tilde{k}_j \right).$$

Given two keys $\tilde{k}_j$, $\tilde{k}_{\bar{j}}$ computed from the same input $x_j$ (i.e., $\tilde{k}_j = \tilde{k}_{\bar{j}}$), the coefficients for these two keys are equivalent $\bar{\alpha}_{i,j} = \bar{\alpha}_{i,\bar{j}}$ and thus the coefficients have no positional information, i.e., the coefficients cannot be discriminated depending on the index $j$.

Now, consider the case $A_t \neq \mathbb{I}_n, \forall t$, then

$$\alpha_{i,j} = \phi \left( q_i^\top \left( \prod_{t=j+1}^{i} A_t \right) b_j k_j \right).$$

Using the same simplification for $b_j$ as above and let $A_{i,j} = \prod_{t=j+1}^{i} A_t$, we get

$$\bar{\alpha}_{i,j} = \phi \left( q_i^\top A_{i,j} \tilde{k}_j \right).$$

The evolution $A_{i,j}$ contains information about the (time) index evolution from $j$ to $i$, since it defines a map $f : \mathbb{R}^n \to \mathbb{R}^n$ that maps any vector $a_j \in \mathbb{R}^n$ to $a_i = A_i \cdots \cdots A_{j+1} a_j$, i.e., describes the evolution of $a_j$ from (time) index $j$ to $i$. Consider again two keys $\tilde{k}_j$, $\tilde{k}_{\bar{j}}$ computed from the same input $x_j$ (i.e., $\tilde{k}_j = \tilde{k}_{\bar{j}}$), then their respective coefficients are not equivalent due to $A_{i,j}$:

$$\bar{\alpha}_{i,j} = \phi \left( q_i^\top A_{i,j} \tilde{k}_j \right) \neq \phi \left( q_i^\top A_{i,\bar{j}} \tilde{k}_{\bar{j}} \right) = \bar{\alpha}_{i,\bar{j}}.$$

Specifically, if $\bar{j} > j$, then the following relation holds

$$A_{i,j} = \prod_{t=j+1}^{i} A_t = \prod_{t=\bar{j}+1}^{i} A_t \prod_{t=j+1}^{\bar{j}} A_t = A_{i,\bar{j}} A_{\bar{j},j},$$

and thus there exists a linear relation between $A_{i,j}$ and $A_{i,\bar{j}}$. Therefore, the coefficients have positional information, quantified by $A_{\bar{j},j}$. $\qquad\square$

**Remark 3.** *The proof of Lemma 3 relies on the fact that keys and parameters such as $b_j$ are computed from a single input $x_j$. However, this is not true if the inputs are preprocessed by a short convolution, i.e., $\{\bar{x}_j\}_{j=1}^i = \Phi \star \{x_j\}_{j=1}^i$, where $\Phi$ denotes the convolution weights. In this case, $\bar{x}_j$ contains scaled copies of previous inputs, depending on the convolution dimension, e.g., $\bar{x}_j = \Phi_0 x_j + \Phi_1 x_{j-1} + \Phi_2 x_{j-2} + \Phi_3 x_{j-3}$. Therefore, preprocessing the inputs with a convolution can offer another way to embed positional information in the coefficients $\alpha_{i,j}$. However, note that this "moving averaging" of inputs might not work for certain repeating input signals.*

### C.1.4 PROOF OF LEMMA 4

**Lemma 4.** *Imposing any of the following structures on $A_i$ in equation 3, results in the corresponding allowed transformations:*

*a) $A_i = \lambda_i \mathbb{I}_n, \lambda_i \in \mathbb{R}, |\lambda_i| \leq 1$; allows scaling of keys (including flipping, if $\lambda_i < 0$) uniformly along all dimensions $n$,*

*b) $A_i = diag(\lambda_i), \lambda_i \in \mathbb{R}^n, |\lambda_i^{(r)}| \leq 1$ where $\lambda^{(r)}$ denotes the $r$-th entry of $\lambda_i$; allows scaling of keys (including flipping, if $\lambda_i^{(r)} < 0$) separately along dimensions $n$,*

*c) $A_i = \mathbb{I}_n - \beta_i \lambda_i \lambda_i^\top, \beta_i \in [0, 2], \lambda_i \in \mathbb{R}^n$, (Householder matrix); allows scaling and specific rotations of keys.*

*Proof.* All three items follow by a direct computation of $A_i$'s action on an arbitrary key vector $k_j \in \mathbb{R}^n$ and by inspecting eigenvalues/eigenvectors.

**(a) Uniform scaling.** If $A_i = \lambda_i \mathbb{I}_n$ then for any key $k_j$ we have

$$A_i k_j = \lambda_i \mathbb{I}_n k_j = \lambda_i k_j,$$

so $A_i$ scales $k_j$ uniformly in every coordinate by the scalar $\lambda_i$. If $|\lambda_i| \leq 1$ the operator is a contraction; if $\lambda_i < 0$ it also flips the sign of every component (a global reflection through the origin).

**(b) Coordinate-wise scaling.** If $A_i = \text{diag}(\lambda_i^{(1)}, \ldots, \lambda_i^{(n)})$, then

$$(A_i k_j)^{(r)} = \lambda_i^{(r)} k_j^{(r)}, \qquad r = 1, \ldots, n.$$

Thus each coordinate is scaled independently by $\lambda_i^{(r)}$. The constraints $|\lambda_i^{(r)}| \leq 1$ render the operation a contraction in each coordinate; negative entries produce sign flips on the corresponding coordinate only.

**(c) Rank-one update / Householder-type transform.** Given $A_i = \mathbb{I}_n - \beta_i \lambda_i \lambda_i^\top$, we decompose any $k_j \in \mathbb{R}^n$ into components parallel and orthogonal to $\lambda_i$:

$$k_j = k_j^{\|} + k_j^{\perp}, \qquad k_j^{\|} = \frac{\lambda_i^\top k_j}{\|\lambda_i\|^2} \lambda_i, \quad \lambda_i^\top k_j^{\perp} = 0,$$

where $\|\cdot\|$ as the Euclidean in $\mathbb{R}^n$. Then

$$A_i k_j = (\mathbb{I}_n - \beta_i \lambda_i \lambda_i^\top)(k_j^{\|} + k_j^{\perp}) = (1 - \beta_i \|\lambda_i\|^2) k_j^{\|} + k_j^{\perp}.$$

Hence $k_j^{\perp}$ (every vector orthogonal to $\lambda_i$) is an eigenvector with eigenvalue 1, and the direction $\lambda_i$ itself is an eigenvector with eigenvalue $1 - \beta_i \|\lambda_i\|^2$. Therefore, $A_i$ acts by scaling the component along $\lambda_i$ by the factor $1 - \beta_i \|\lambda_i\|^2$, while leaving the orthogonal complement unchanged.

Special cases and remarks:

• If one chooses $\beta_i = 2/\|\lambda_i\|^2$ (and $\lambda_i \neq 0$), then $1 - \beta_i \|\lambda_i\|^2 = -1$ and

$$A_i = \mathbb{I}_n - \frac{2}{\|\lambda_i\|^2} \lambda_i \lambda_i^\top$$

is a classical Householder reflection (an orthogonal matrix with determinant $-1$) that reflects across the hyperplane orthogonal to $\lambda_i$.

- For general $\beta_i \in [0,2]$ and $\|\lambda_i\|$ normalized (or when $\beta_i\|\lambda_i\|^2 \in [0,2]$), the eigenvalue $1 - \beta_i\|\lambda_i\|^2$ lies in $[-1,1]$, so the operator is a contraction on the $\lambda_i$-direction and may flip the sign (reflection), if the eigenvalue is negative.

- Although a single rank-one transform of this form does not produce an arbitrary rotation, compositions of Householder reflections (each of which is of the form $\mathbb{I}_n - 2\lambda\lambda^\top$ with unit $\lambda \in \mathbb{R}^n$ can generate any orthogonal matrix (rotations and reflections). Hence, by composing appropriate special cases of the matrices in (c), orthogonal rotations can be realized.

Combining these elementary computations proves that each structural constraint on $A_i$ yields the stated class of allowed transformations. $\square$

**Remark 4.** *In Lemma 4 and the proof above, we make no assumption on the magnitude of $\lambda_i$ in a Householder-type evolution matrix. However, as shown in the proof, assuming unit $\lambda_i$ is necessary to define all transformations. This is the reason why the keys $k_j$ and queries $q_i$ are L2 normalized in Yang et al. (2024; 2025a) for the computation of $A_i$.*

### C.1.5 PROOF OF LEMMA 5

**Lemma 5.** *Consider two i.i.d. normally distributed random vectors $x_i, x_j \sim \mathcal{N}(0, \Sigma)$. Then, the dot product $q_i^\top h_{i,j}$ with $q_i, h_{i,j}$ defined in equation 4, has zero-mean and variance $Var(q_i^\top h_{i,j}) = tr(\Sigma_q \Sigma_h)$ with $\Sigma_q = T_q \Sigma T_q^\top$, $\Sigma_h = T_{h,i} \Sigma T_{h,i}^\top$. Assuming $\Sigma = \sigma \mathbb{I}_d$, both $\Sigma_q$ and $\Sigma_h$ are positive semi-definite and the variance of the dot product scales as $Var(q_i^\top h_{i,j}) = \mathcal{O}(n)$.*

*Proof.* We proof the lemma in three parts, first zero mean, then the variance relation, and finally the scaling under the isotropy assumption.

**Mean.** Since $x_i$ and $x_j$ are independent and zero-mean,

$$\mathbb{E}[q_i^\top h_{i,j}] = \mathbb{E}[(T_q x_i)^\top (T_{h,i} x_j)] = \mathbb{E}_{x_i} \mathbb{E}_{x_j}[x_i^\top T_q^\top T_{h,i} x_j] = 0,$$

hence the mean is zero.

**Variance.** Since the mean is zero,

$$\mathrm{Var}(q_i^\top h_{i,j}) = \mathbb{E}\big[(q_i^\top h_{i,j})^2\big] = \mathbb{E}\big[(T_q x_i)^\top (T_{h,i} x_j)(T_{h,i} x_j)^\top (T_q x_i)\big].$$

Conditioning on $x_i$ and taking expectation over $x_j$ first,

$$\mathbb{E}_{x_j}\big[(T_{h,i} x_j)(T_{h,i} x_j)^\top\big] = T_{h,i} \mathbb{E}[x_j x_j^\top] T_{h,i}^\top = T_{h,i} \Sigma T_{h,i}^\top := \Sigma_h,$$

and due to symmetry $\mathbb{E}_{x_i}\big[(T_q x_i)(T_q x_i)^\top\big] = T_q \Sigma T_q^\top = \Sigma_q$. Therefore,

$$\mathbb{E}\big[(q_i^\top h_{i,j})^2\big] = \mathbb{E}_{x_i}\big[(T_q x_i)^\top \Sigma_h (T_q x_i)\big] = \mathbb{E}_{x_i}\big[\mathrm{tr}\big(\Sigma_h (T_q x_i)(T_q x_i)^\top\big)\big] = \mathrm{tr}\big(\Sigma_q \Sigma_h\big),$$

due to the previous relations and the cyclic property of the trace operator.

**Isotropy.** If $\Sigma = \sigma \mathbb{I}_d$, then

$$\Sigma_q = \sigma T_q T_q^\top, \qquad \Sigma_h = \sigma T_h T_h^\top,$$

which are symmetric positive semidefinite by definition of matrix outer products (Friedberg et al., 1997). Therefore,

$$\mathrm{Var}(q_i^\top h_{i,j}) = \mathrm{tr}\big(\Sigma_q \Sigma_h\big) = \sigma^2 \mathrm{tr}\big(T_q T_q^\top T_{h,i} T_{h,i}^\top\big) = \sigma^2 \|T_q^\top T_{h,i}\|_F^2,$$

where we used the cyclic property of the trace and the identity $\|M\|_F = \mathrm{tr}(MM^\top)$ (Friedberg et al., 1997), with $\|\cdot\|_F$ denoting the Frobenius norm. To prove the scaling statement, we assume $T_q, T_{h,i}$ are properly normalized weight matrices (i.e. the matrices have entries $\mathcal{O}(1/\sqrt{n})$, such that the output variance remains $\mathcal{O}(1)$). This is assumption is common in practice. Therefore, we have $\|T_q\|_F = \mathcal{O}(\sqrt{n})$, $\|T_{h,i}\|_F = \mathcal{O}(\sqrt{n})$ and thus $\|T_q^\top T_{h,i}\|_F^2 = \mathcal{O}(n)$ and $\mathrm{Var}(q_i^\top h_{i,j}) = \mathcal{O}(n)$. Importantly, given common normalization choices for $T_q, T_{h,i}$, which keep per-coordinate variances to $\mathcal{O}(1)$, the dot product variance grows linearly in $n$. $\square$

### C.1.6 PROOF OF LEMMA 6

**Lemma 6.** *Consider a fixed index $j$ and let $s_i := q_i^\top h_{i,j}$, $\alpha_i = \phi(s_i)$ in equation 3. Let $\phi : \mathbb{R} \to [0, \infty)$ be nondecreasing and unbounded, and let $g : [0, \infty) \to (0, \infty)$ be an increasing and unbounded comparison function, such that $L := \limsup_{i \to \infty} \frac{\phi(s_i)}{g(s_i)} < \infty$. If $\eta_i$ is chosen such that $\liminf_{i \to \infty} \frac{\eta_i}{g(s_i)} := m > 0$, then the normalized coefficients are bounded: $\limsup_{i \to \infty} \frac{\alpha_i}{\eta_i} \le \frac{L}{m}$.*

*Proof.* Fix $\varepsilon \in \big(0, \min\{1, m\}\big)$. By the definition of the limit superior, there exists $N_1 \in \mathbb{N}$ such that for all $i \ge N_1$,

$$\frac{\phi(s_i)}{g(s_i)} \le L + \varepsilon.$$

By the definition of the limit inferior, there exists $N_2 \in \mathbb{N}$ such that for all $i \ge N_2$,

$$\frac{\eta_i}{g(s_i)} \ge m - \varepsilon.$$

Hence for all $i \ge N := \max\{N_1, N_2\}$ we have

$$\frac{\alpha_i}{\eta_i} = \frac{\phi(s_i)}{\eta_i} = \frac{\phi(s_i)}{g(s_i)} \cdot \frac{g(s_i)}{\eta_i} \le \frac{L + \varepsilon}{m - \varepsilon}.$$

Since finitely many initial indices do not affect the limit superior, letting $\varepsilon \to 0$ yields

$$\limsup_{i \to \infty} \frac{\alpha_i}{\eta_i} \le \frac{L}{m},$$

which proves the lemma. $\qquad\square$

**Remark 5.** *Assuming the normalization factor $\eta_i$ is a function of the coefficients themselves, i.e., $\eta_i = f(\alpha_{i,j})$, simplifies Lemma 6 significantly, since we only need to analyze $\frac{\alpha_i}{f(\alpha_i)}$ (without comparison function $g$). Hence, if $f(\cdot)$ grows linearly, $\alpha_i/f(\alpha_i)$ converges to a constant $m > 0$; if $f(\cdot)$ grows superlinearly, $\alpha_i/f(\alpha_i)$ converges to $0$; and $f(\cdot)$ grows sublinearly, $\alpha_i/f(\alpha_i)$ blows up.*

### C.2 COROLLARIES

#### C.2.1 PROOF OF COROLLARY 2.1

**Corollary 2.1.** *Consider the kernel approximation of $\phi(\cdot) : \mathbb{R}^n \to \mathbb{R}^q$ to be $\tilde{\phi}(q_i^\top h_{i,j}) = \psi_q(q_i)^\top \psi_h(h_{i,j})$, with $\psi_q : \mathbb{R}^n \to \mathbb{R}^q$ and $\psi_h : \mathbb{R}^n \to \mathbb{R}^q$, such that it approximates the readout map $\phi(\cdot) \approx \tilde{\phi}(\cdot)$ in equation 3. Then, the zero-level set of the approximation $\tilde{\phi}(\cdot)$ is the singleton $\mathcal{Z} = \{0\}$ and by Lemma 2 the set of linear transformations $\mathcal{T}$ is small.*

*Proof.* Given that the kernel approximation is defined as $\tilde{\phi}(q_i^\top h_{i,j}) = \psi_q(q_i)^\top \psi_h(h_{i,j})$, we can rewrite this approximation using a change of variables as

$$\tilde{\phi}(q_i^\top h_{i,j}) = \tilde{q}_i^\top \tilde{h}_i, j, \quad \text{with } \tilde{q}_i := \psi_q(q_i), \; \tilde{h}_{i,j} := \psi_h(h_i, j). \tag{15}$$

Note that equation 15 is equivalent to a dot product with an identity readout map $\text{Id}(\cdot)$, albeit in different dimensions than the untransformed dot product $q_i^\top h_{i,j}$ ($\tilde{q}_i \in \mathbb{R}^q$ compared to $q_i \in \mathbb{R}^n$). Given that the zero-level set of a dot product is the singleton $\{0\}$, the zero-level set of $\tilde{\phi}(\cdot)$ is $\mathcal{Z} = \{0\}$. Hence, $|\mathcal{Z}| = 0$ and the measure of the set of linear transformations reduces to $|\mathcal{T}| = g(\{0\})$ (see proof of Lemma 2 for the exact definitions). Therefore, $\mathcal{T}$ is reduced to a single fiber, i.e., $\mathcal{T} = \{(T_q, T_{h,i}) \mid (T_q x_i)^\top (T_{h,i} x_j) = 0\}$. $\qquad\square$

#### C.2.2 PROOF OF COROLLARY 2.2

**Corollary 2.2.** *Let $y_L \in \mathbb{R}^{d_v}$ be the solution to equation 2, where $\alpha_{L,j} : \mathbb{R}^n \to \mathbb{R}$; $q_L \mapsto \alpha_{L,j}(q_L)$ is defined in equation 3 with identity readout map and normalization factor, i.e., $\phi(\cdot) = \text{Id}(\cdot)$, $\eta_L = 1$. Consider the set of linearly independent states $\mathcal{H} = \{h_{L,t} \mid c_1 h_{L,1} + \cdots + c_t h_{L,t} = 0 \implies c_1 = \cdots = c_t = 0\}$. Then, a nonzero $q_L$ that achieves $\alpha_{L,j} = 0$, $\forall h_{L,j} \in \mathcal{H}$ exists if and only if $\dim\big(\text{span}\{h_{L,j} \in \mathcal{H}\}\big) < n$. In particular, given a nonzero $q_L$, the measure $|\mathcal{H}| \le n - 1$.*

*Proof.* The conditions $\alpha_{L,j} = q_L^\top h_{L,j} = 0$, $\forall h_{L,j} \in \mathcal{H}$ are equivalent to the linear homogeneous system

$$H^\top q_L = 0, \tag{16}$$

where $H \in \mathbb{R}^{n \times |\mathcal{H}|}$ with $|\mathcal{H}|$ the measure of set $\mathcal{H}$,[18] is a matrix whose columns consist of the vectors $h_{L,j} \in \mathcal{H}$. The solution space for $q_L$ is the nullspace of $H^\top$. By the rank–nullity theorem (Friedberg et al., 1997, Theorem 2.3), we get

$$\dim\big(\ker(H^\top)\big) = n - \text{rank}(H^\top) = n - \text{rank}(H).$$

Hence a nonzero $q_L$ solving equation 16 exists if and only if $\text{rank}(H) < n$, i.e., if and only if the vectors $h_{L,j} \in \mathcal{H}$ span a proper subspace of $\mathbb{R}^n$.

Consequently, the largest possible number of linearly independent constraints (equivalently, independent $h_{L,j}$) that admit a nonzero solution is $n - 1$. Equivalently, at most $n - 1$ independent coefficients $\alpha_{L,j}$ can be forced to zero simultaneously, with a nonzero choice of $q_L$. $\qquad\square$

### C.2.3 PROOF OF COROLLARY 6.1

**Corollary 6.1.** *Choosing $\eta_i = \sum_{j=1}^i \alpha_{i,j}$ in equation 2 constrains the normalized coefficients to $\tilde{\alpha}_{i,j} = \alpha_{i,j}/\eta_i \in [0, 1]$ and imposes $\sum_{j=1}^i \tilde{\alpha}_{i,j} = 1$.*

*Proof.* Consider the linear combination (equation 2)

$$y_i = \sum_{j=1}^i \frac{\alpha_{i,j}}{\eta_i} v_j.$$

Then, setting $\eta_i = \sum_{j=1}^i \alpha_{i,j}$, results in

$$y_i = \frac{\sum_{j=1}^i \alpha_{i,j} v_j}{\sum_{j=1}^i \alpha_{i,j}} = \sum_{j=1}^i \tilde{\alpha}_{i,j} v_j,$$

with $\tilde{\alpha}_{i,j} = \alpha_{i,j}/\eta_i$ as used in the statement. Therefore, the coefficients $\tilde{\alpha}_{i,j}$ trivially sum to 1. Additionally, this restricts the coefficients and specializes the linear combination to a convex combination (if $\tilde{\alpha}_{i,j} \geq 0$) or an affine combination, as discussed in Appendix A. Therefore, the output $y_i$ lies in either the simplex or a hyperplane spanned by the value vectors $v_j$ (Friedberg et al., 1997). $\qquad\square$

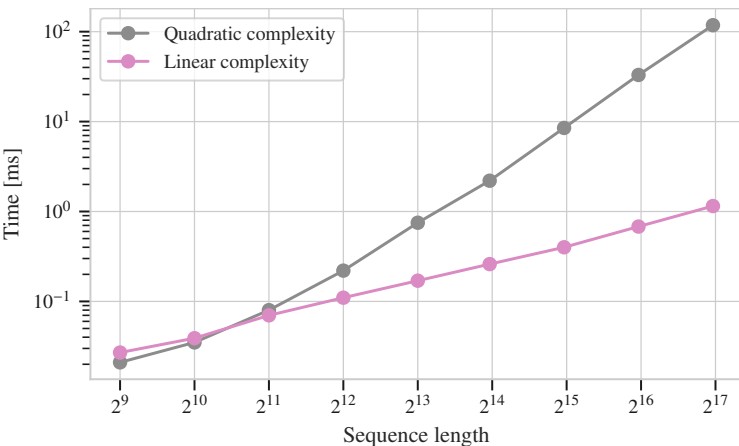

Figure 5: **(Principle 1)** Computation time of a recurrent (magenta) and non-recurrent (gray) implementation against sequence length.

---

[18]This means the number of vectors $h_{L,j}$ contained in the set.

## D EXTENDED EXPERIMENTAL VALIDATION

In this appendix, we provide additional experiments to validate the principles stated in the main text.

### D.1 PRINCIPLE 1

Since Principle 1 and Lemma 1 are existence results, the principle cannot be validated experimentally. However, the principle has profound implications on the efficiency of sequence models as shown in Figure 5. With respect to sequence length, a recurrent implementation has linear complexity, while a non-recurrent implementation has quadratic complexity. Figure 5 shows the computation time of FlashAttention-2 (Dao, 2023) – a non-recurrent implementation – and SSD (Dao & Gu, 2024) – a reccurent implementation – against sequence length.[19]

### D.2 PRINCIPLE 2 AND SUB-PRINCIPLES

Figure 6 adds to Figure 2 in the main text, by adding the memorization task and additionally showing the behavior of the investigated readout maps $\phi(\cdot)$ close to zero, which highlights how the theoretical near-zero sets are computed.

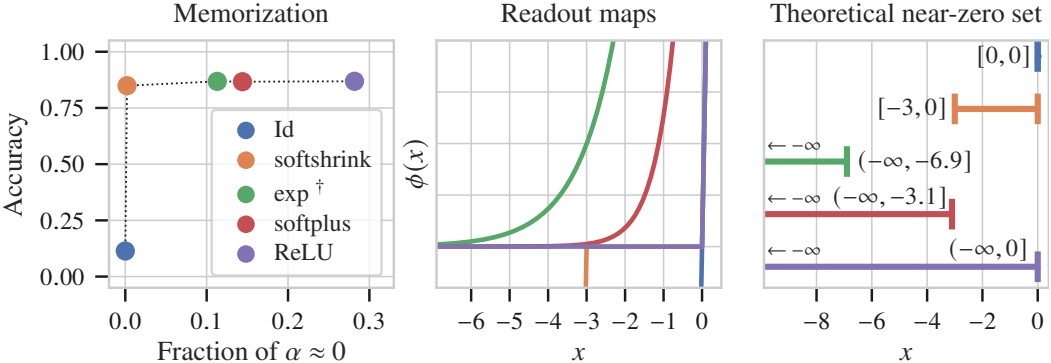

Figure 6: **(Principle 2)** Performance of different readout maps $\phi(\cdot)$ on the memorization task of MAD against the fraction of coefficients with near zero values ($|\alpha| \leq 0.001$; left). The other parameters $A_i, b_j, \eta_i$ are fixed, thus the setting for $^\dagger$ is equivalent to softmax attention. Behavior of the investigated readout maps $\phi(\cdot)$ close to zero (middle) and the theoretical near-zero sets of each readout map ($|\phi(x)| \leq 0.001$; right).

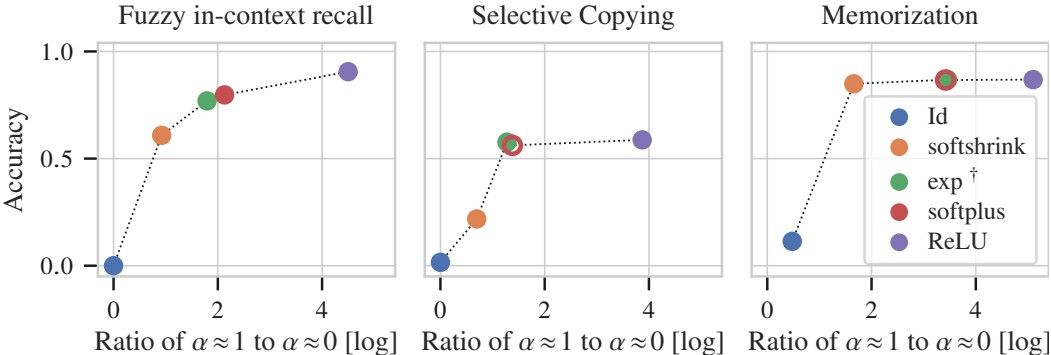

Figure 7: **(Principle 2)** Performance of different readout maps $\phi(\cdot)$ on three MAD tasks with respect to the ratio of coefficients $\alpha_{i,j}$ set to large ($\approx 1$) and near zero values ($\approx 0$). The other parameters $A_i, b_j, \eta_i$ are fixed, thus the setting for $^\dagger$ is equivalent to softmax attention.

---

[19]The data for this plot was obtained using `https://github.com/state-spaces/mamba`.

Figure 7 additionally validates Principle 2 by showing performance against the ratio of coefficients $\alpha_{i,j}$ set to large values ($\alpha \in [0.9, 1]$) and coefficients contained in the near-zero set ($|\alpha| \leq 0.001$); since the ratio is small, we plot it on a log-scale. This ratio captures not only a readout map's ability to suppress coefficients, but also its ability to select coefficients close to $1$. *The results in Figure 7 suggest that as the ability of a readout map to set coefficients to zero increases, the model becomes more selective (also setting more coefficients to* 1*), increasing the performance of all three tasks.*

**Principle 2.1**   To validate Principle 2.1, we ablate four kernel approximation maps $\psi(\cdot) \in \{\text{ReLU}(\cdot), \text{ELU}(\cdot) + 1, \text{softplus}(\cdot), \exp(\cdot)\}$ that approximate the readout map as $\phi(\cdot) \approx \psi(\cdot)\psi(\cdot)$. The other coefficient dynamics parameters are fixed to $A_i = \mathbb{I}_n$, $b_j = 1/\sqrt{n}$, and $\eta_i = \sum_j \alpha_{i,j}$. In Figure 8, we report the performance of all approximation maps on the fuzzy in-context recall and selective copying tasks of MAD, against the fraction of coefficients in the near-zero set (equivalent to Figure 2). Additionally, we provide the performance of softmax attention ($\phi(\cdot) = \exp(\cdot)$) and pure linear attention ($\phi(\cdot) = \text{Id}(\cdot)$) as baselines. *As dictated by the principle, the kernel approximations set a smaller fraction of coefficients in the near-zero set – compared to the readout maps in Figure 2 – thus deteriorating performance on both tasks.* However, note that most kernel approximations ($\psi(\cdot) \in \{\text{ELU}(\cdot) + 1, \text{softplus}(\cdot), \exp(\cdot)\}$) are more efficient at setting coefficients close to zero than the identity readout map.

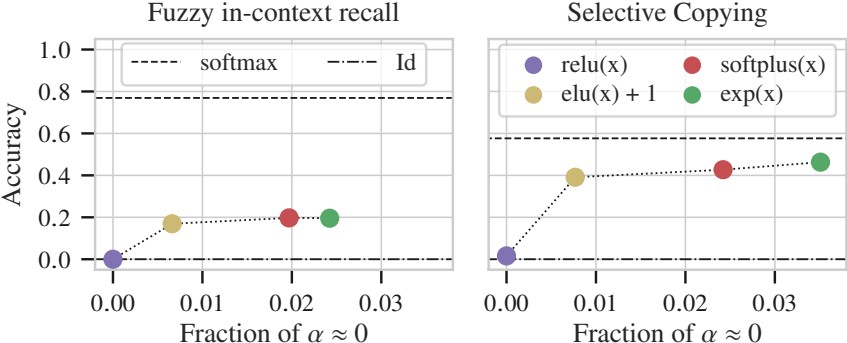

Figure 8: **(Principle 2.1)** Performance of different kernel approximations $\psi(\cdot)$ of the readout map $\phi(\cdot) \approx \psi(\cdot)\psi(\cdot)$ on two MAD tasks against the fraction of coefficients with near zero values ($|\alpha| \leq 0.001$). The other parameters $A_i$, $b_j$, $\eta_i$ are fixed. Dotted lines indicate the performance level of softmax attention ($\phi(\cdot) = \exp(\cdot)$) and pure linear attention ($\phi(\cdot) = \text{Id}(\cdot)$) from Figure 2, as a baseline.

**Principle 2.2**   To validate Principle 2.2, we investigate the number of coefficients that are set to zero per time index $i$. Since no coefficient $\alpha_{i,j}$ is exactly set to zero due to numerical precision, we treat $|\alpha_{i,j}| \leq 1e{-}6$ as zero. Using two models trained for Figure 8 (Principle 2.1), i.e., readout approximations $\psi(\cdot) \in \{\text{softplus}(\cdot), \exp(\cdot)\}$, we compute for each time index $i$ and each head, the average number of zero coefficients over a sample input batch of size $64$. Since the two models have state dimension $n = 128$ and 16 heads, the effective state dimension[20] is $\tilde{n} = \frac{n}{16} = 8$, therefore we expect no more than $\tilde{n} - 1 = 7$ zero coefficients per time index and head (Principle 2.2). The results are shown in Figure 9 for one example head of each model on both tasks. *The figure shows that the allowed number of zero coefficients is never exceeded and in general longer sequence lengths allow to set more coefficients to zero (compare the two tasks with different sequence lenghts).* This can be explained by the *linear independence* of states $h_{i,j}$ required by Corollary 6.1: for longer sequences, there exist more linearly independent states, which facilitates setting more coefficients to zero.

---

[20]In a multi-head setting, the state dimension $n$ in Principle 2.2 needs to be replaced by the head dimension $\tilde{n} = n/\text{nr. of heads}$.

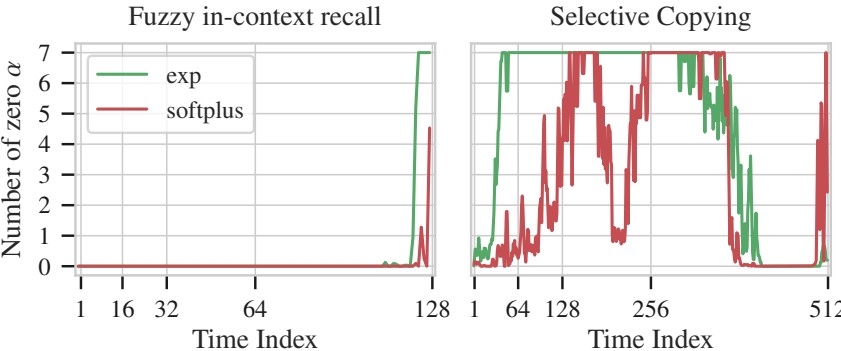

Figure 9: **(Principle 2.2)** Number of zero coefficients per time index $i$, for one head of two models $\psi(\cdot) \in \{\text{softplus}(\cdot), \exp(\cdot)\}$ trained on fuzzy in-context recall and selective copying (same models as in Fig. 8).

## D.3 PRINCIPLE 5

To validate Principle 5, we ablate four choices of scaling factors $b_j$, on four combinations of readout maps $\phi(\cdot)$ and evolution matrices $A_i$. We choose two scaling factors that adhere to the principle ($b_j = 1/\sqrt{n}$, $b_j = z_j/\sqrt{n}$) and two that do not ($b_j = 1$, $b_j = z_j$), where $z_j$ is input-dependent, i.e., $z_j = \sigma(W_z x_j)$ with $\sigma(\cdot)$ the sigmoid function. For readout maps and evolution matrices, we choose the combinations $(\phi(\cdot), A_i) = \{(\exp(\cdot), \mathbb{I}_n), (\text{ReLU}(\cdot), \mathbb{I}_n), (\text{Id}(\cdot), \alpha_i \mathbb{I}_n), (\text{Id}(\cdot), \mathbb{I}_n - \beta_i k_i k_i^\top)\}$, where $\alpha_i$ is parameterized equivalently to $\exp(-\Delta_i A)$ in Mamba-2 (Dao & Gu, 2024) and $\beta_i$ is input-dependent, i.e., $\beta = 2\sigma(W_\beta x_i)$ with $\sigma(\cdot)$ the sigmoid function. These models are trained on the fuzzy in-context recall task of MAD and the results are shown in Figure 10. We plot the accuracy on the task, across three different seeds, against the effective head dimension, i.e., $n/\text{nr. of heads}$.[21] *The results in Figure 10 suggest that for scaling factors not adhering to the principle ($b_j = 1$, $b_j = z_j$), the performance depends on the random seed, as we notice large performance fluctuations. Conversely, for scaling factors adhering to the principle ($b_j = 1/\sqrt{n}$, $b_j = z_j/\sqrt{n}$), the performance remains largely constant across seeds.* This behavior is expected from Principle 5, since the dot product $q_i^\top h_{i,j}$ has larger variance for $b_j$ not adhering to the principle. Note that the performance fluctuations are largest for readout map $\phi(\cdot) = \exp(\cdot)$. We hypothesize that this is due to the exponential growth of the readout map, as dot products $q_i^\top h_{i,j}$ with large variance are either pushed to zero or large values. This is necessarily less pronounced for the other three combinations, although performance fluctuations are still more pronounced for scaling factors not adhering to the principle. While there are indications that performance fluctuates more as the head dimension increases (see e.g. "scalar" for $b_j = 1$) – which is expected per Principle 5 – this pattern is not consistent. We hypothesize that the head dimension is too small for this pattern to emerge consistently.

Table 3: Perplexity of sequence models with different choices of scaling factor $b_j$ pretrained on Wikitext-103.

|  | Mamba-2 | | | Softmax Attention | |
|---|---|---|---|---|---|
| scaling parameter | $b_j = 1$ | $b_j = 1/\sqrt{n}$ | $b_j = \Delta_j$ | $b_j = 1$ | $b_j = 1/\sqrt{n}$ |
| parameter count | 110 M | 110 M | 110 M | 120 M | 120 M |
| perplexity | 29.23 | 28.84 | 28.87 | 32.69 | 31.73 |

We also validate Principle 5 on a more realistic dataset than the previous synthetic one. For this, we train a standard softmax attention model and a standard Mamba-2 model on Wikitext-103 (Merity et al., 2016). For both models, we modify the scaling factor $b_j$ and fix all other parameters to the standard choices of each model. Both models, have the same size but softmax attention is equipped

---

[21]In a multi-head setting, the state dimension $n$ in $1/\sqrt{n}$ is replaced by the head dimension $\tilde{n} = n/\text{nr. of heads}$.

with additional positional embeddings, which explains the higher parameter count in Table 3. The results in Table 3 suggest that there exists a performance gap, yet small, between scaling factors adhering to the principle ($b_j = 1/\sqrt{n}$, $b_j = \Delta_j$)[22] and scaling factors that do not ($b_j = 1$). The finding that softmax attention without scaling of the keys can perform well in practice, has also been observed in Britz et al. (2017). Finally, note that decoupling the parameters $A_i$ and $b_j$ in Mamba-2 does not hurt performance, with the scaling factor $b_j = 1/\sqrt{n}$ (with $\Delta_i$ still used to compute $A_i$) performing on par with $b_j = \Delta_j$.

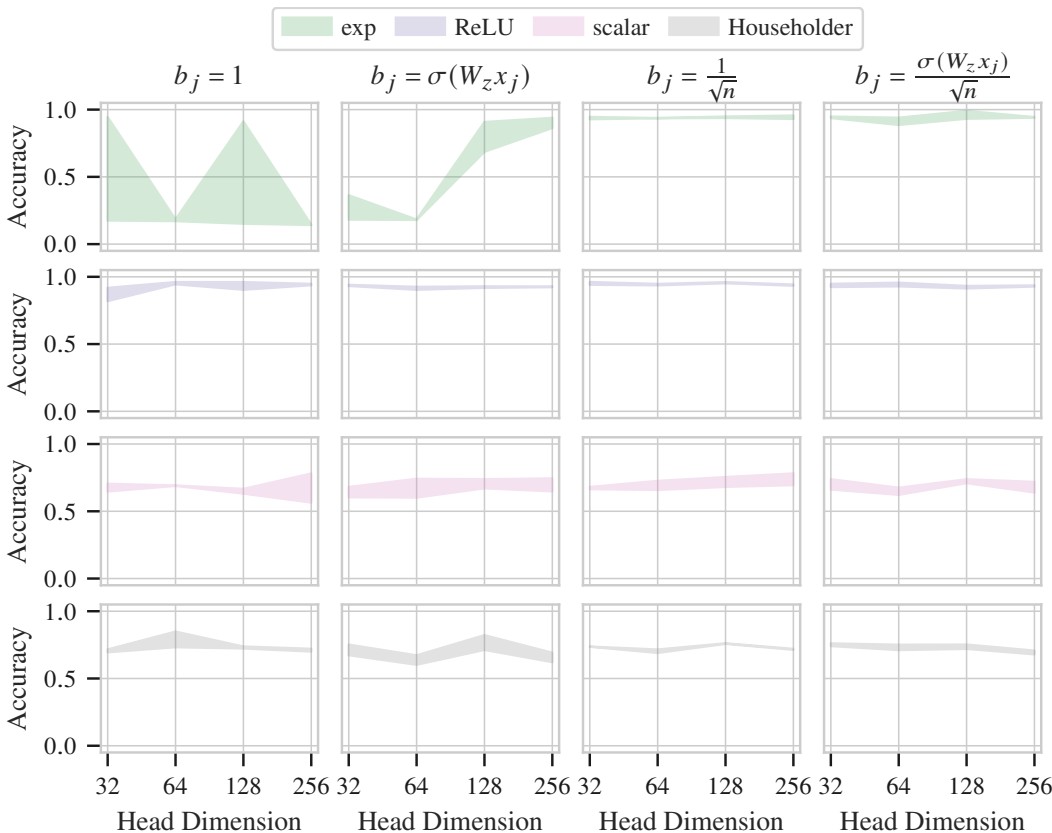

Figure 10: **(Principle 5)** Minimal and maximal performance across three seeds on the fuzzy in-context recall task, for different choices of scaling factor $b_j$ and combinations $(\phi(\cdot), A_i)$. The legend refers to the combinations $(\phi(\cdot), A_i) = \{(\exp(\cdot), \mathbb{I}_n), (\text{ReLU}(\cdot), \mathbb{I}_n), (\text{Id}(\cdot), \alpha_i \mathbb{I}_n), (\text{Id}(\cdot), \mathbb{I}_n - \beta_i k_i k_i^\top)\}$ and each column shows the results for a single $b_j$.

### D.4 PRINCIPLE 6

Principle 6 is mainly concerned with training stability related to the evolution matrices $A_i$ and normalization factors $\eta_i$. Due to Corollary 6.1, choosing $\eta_i = \sum_j \alpha_{i,j}$ guarantees stable training and has additional implications on the output space of the sequence model (see Appendix A). We hence focus on evolution matrices to experimentally validate the principle. It is widely known, that stable $A_i$ (all eigenvalues lie in the unit circle of the complex plane) guarantee numerically stable training, as evidenced by the design of $A_i$ in e.g. S4 (Gu et al., 2022a), LRU (Orvieto et al., 2023), Mamba-2 (Dao & Gu, 2024), GLA (Yang et al., 2023), and DeltaNet (Schlag et al., 2021). To validate the principle, we therefore consider the simple unstable evolution matrix $A_i = 1.05 \mathbb{I}_n$ and show that with appropriate design of $\eta_i$, we are able to solve noisy in-context recall. We fix $b_j = 1$, $A_i = 1.05 \mathbb{I}_n$ and ablate the readout maps $\phi(\cdot) \in \{\exp(\cdot), \text{softplus}(\cdot), \text{ReLU}(\cdot)\}$ and normalization factors $\eta_i \in \{1, 1.05^i, \sum_j \alpha_{i,j}\}$; results are shown in Figure 11. *As expected by Corollary 6.1, the normalization factor $\eta_i = \sum_j \alpha_{i,j}$ ensures stable training and achieves perfect accuracy.* The results of the other two choices need additional discussion.

---

[22]In Mamba-2, $\Delta_j$ adheres to the principle as discussed in Example 2.

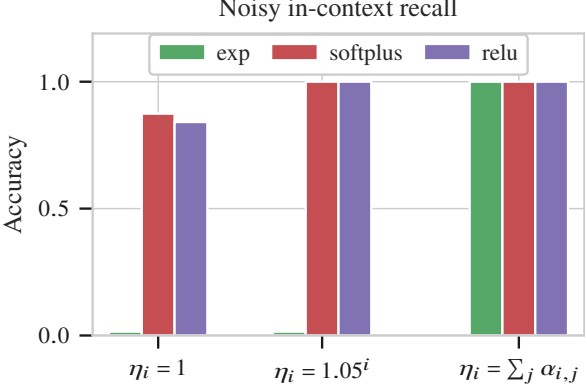

Figure 11: **(Principle 6)** Performance of three normalization factor choices $\eta_i$ on the noisy in-context recall task, for unstable evolution matrix $A_i = 1.05\,\mathbb{I}_n$, $b_j = 1$, and readout maps $\phi(\cdot) \in \{\exp(\cdot), \text{softplus}(\cdot), \text{ReLU}(\cdot)\}$.

First, consider $\eta_i = 1$: for readout map $\exp(\cdot)$, training is unstable and the model fails to learn anything, yet for readout maps $\text{softplus}(\cdot), \text{ReLU}(\cdot)$ training is stable and the model achieves good performance, albeit suboptimal. This is explained by the growth of the readout maps, while $\exp(\cdot)$ grows superlinearly, $\text{softplus}(\cdot), \text{ReLU}(\cdot)$ grow linearly, thus the coefficients $\alpha_{i,j}$ grow faster for $\phi(\cdot) = \exp(\cdot)$ than the other two maps, as the state $h_{i,j}$ grows due to $\prod_{t=j+1}^{i} 1.05\,\mathbb{I}_n$ (see equation 4). Since the task has short sequence length (128) and the chosen evolution matrix is mildly unstable (1.05), readout maps with linear growth can still achieve good performance.

Now consider $\eta_i = 1.05^i$: for readout map $\exp(\cdot)$, training is unstable and the model fails to learn anything, yet for readout maps $\text{softplus}(\cdot), \text{ReLU}(\cdot)$ training is stable and the model achieves perfect performance. This is again explained by the growth of the readout maps, when considering the normalized coefficients $\frac{\alpha_{i,j}}{\eta_i}$. For $\phi(\cdot) = \exp(\cdot)$, the coefficients $\alpha_{i,j}$ grow faster than the normalization factors $\eta_i$, leading to instability. For the other two readout maps, the coefficients $\alpha_{i,j}$ grow as fast as the normalization factors $\eta_i$, thus stabilizing training. Further experimental validation of the principle, is given by the experimental results in Beck et al. (2024), since the model could not have been trained without proper design of the normalization factors $\eta_i$.

## E  DETAILS ON EXPERIMENTAL SETUP

The experimental results provided in Section 5 and Appendix D are performed on the mechanistic architecture design (MAD) benchmark (Poli et al., 2024). To perform the experiments, we modified the MAD code base[23] and integrated it in our existing code base, which will be released upon publication. For pretraining in Appendix D.3, we used the Wikitext-103 (Merity et al., 2016) dataset. All experiments were run on a cluster equipped with 8 NVIDIA RTX 4090 GPUs.

### E.1  GLOBAL SETTINGS

**Wikitext**  The Wikitext-103 dataset is tokenized using the standard GPT-2 tokenizer (`vocab_size`: 50257) and chunked into blocks of sequence length 1024, before being processed by the model.

**MAD Tasks**  We perform experiments on the following four tasks of MAD: selective copying, memorization, noisy in-context recall, and fuzzy in-context recall. For a detailed exposition of these tasks, we refer to Poli et al. (2024, Section 3.1). Below, we report the dataset parameters used for our experiments; we refer to Poli et al. (2024, Appendix B) for a detailed explanation of each parameter and its effect on the task difficulty.

---

[23]https://github.com/athms/mad-lab

- **Selective Copying**: `vocab_size`: 64, `seq_len`: 512, `num_tokens_to_copy`: 32, `selective`: True, `num_train_examples`: 3200.

- **Memorization**: `vocab_size`: 4096, `seq_len`: 32, `frac_noise`: 0.0 (no noise), `num_train_examples`: 256.

- **Noisy In-context Recall**: `vocab_size`: 32, `seq_len`: 128, `multi_query`: True, `frac_noise`: 0.2 (noisy), `noise_vocab_size`: 16, `num_train_examples`: 3200.

- **Fuzzy In-context Recall**: `vocab_size`: 32, `seq_len`: 128, `multi_query`: True, `frac_noise`: 0.0 (no noise), `num_train_examples`: 12800.

**Training**   For MAD tasks, we use the parameters shown in Table 4. These are the same parameters as proposed in Poli et al. (2024), but we sweep a larger range of learning rates. All performances/accuracies reported in this paper, correspond to the best performance/accuracy across the learning rate and weight decay sweep reported in Table 4.

Table 4: Training parameters used on the MAD benchmark.

| | |
|---|---|
| Optimizer | `AdamW` |
| Optimizer Momentum | $(\beta_1, \beta_2) = (0.9, 0.98)$ |
| Dropout | `None` |
| Batch Size | 128 |
| Training Epochs | 200 |
| Learning Rate Schedule | `Cosine Decay` |
| Learning Rate Warmup | `None` |
| Number of Test Samples | 1280 |
| Learning Rate | $[0.0001, 0.0005, 0.001, 0.005, 0.01]$ |
| Weight Decay | $[0.0, 0.1]$ |

For pretraining on Wikitext, we use the training parameters stated in Table 5.

Table 5: Training parameters used for pretraining on Wikitext.

| | |
|---|---|
| Optimizer | `AdamW` |
| Optimizer Momentum | $(\beta_1, \beta_2) = (0.9, 0.95)$ |
| Dropout | `None` |
| Batch Size | 8 |
| Training Steps | 130k |
| Learning Rate Schedule | `Cosine Decay` |
| Warmup Steps | 3k |
| Learning Rate | 0.0006 |
| Weight Decay | 0.1 |

**Models**   The model parameters for each experiment are reported individually in the following section. We use the two block designs shown in Figure 12. Generally, we use the Transformer style block for attention-based models (i.e. readout map $\phi(\cdot) \neq \text{Id}(\cdot)$) and the Gated DeltaNet style block for SSMs/RNNs (i.e. readout map $\phi(\cdot) = \text{Id}(\cdot)$). Additionally, we typically use interleaved MLP blocks after each block (either Type 1 or 2), which do not count towards the number of layers. For

example, a two layer model with interleaved MLP blocks, would effectively consist of four layers, two sequence mixing layers and two MLP layers. The MLP block consists of two dense layers with SwiGLU activation (Shazeer, 2020) and inner dimension reported per experiment.

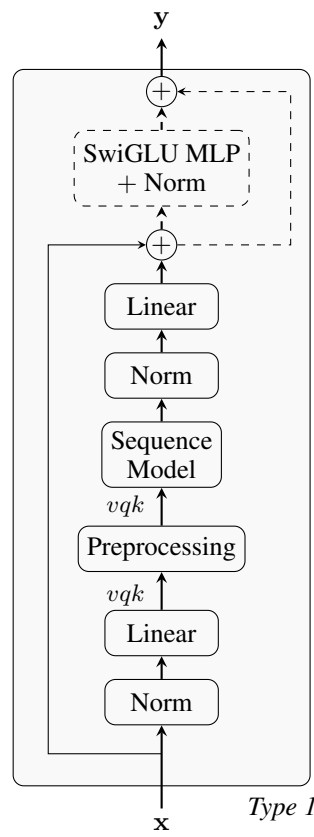 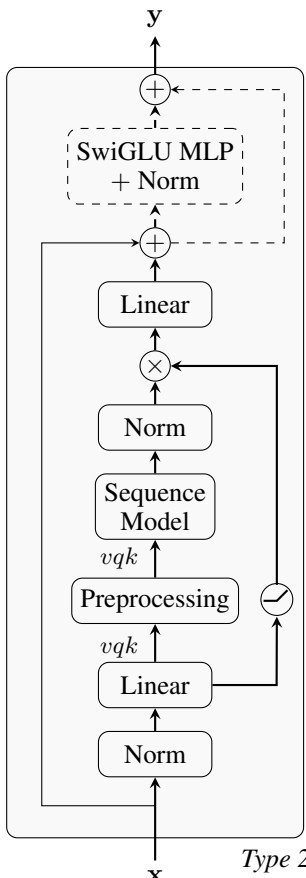

Figure 12: Block designs used in experiments: Transformer style (left; Type 1) and Gated DeltaNet style (right, Type2) from (Yang et al., 2025a, Fig. 1). Dashed parts show the optional SwiGLU MLP.

### E.2 MODEL PARAMETERS PER EXPERIMENT

In the following, we detail the model parameters used to experimentally validate each principle individually.

- The model parameters used to obtain Figures 2, 6 & 7 and validate Principle 2 are provided in Table 6.
- The model parameters used to obtain Figures 8 & 9 and validate Principles 2.1 and 2.2 are provided in Table 7.
- The model parameters used to obtain Figure 1 and validate Principle 3 are provided in Table 8.
- The model parameters used to obtain Figure 3 and validate Principle 4 are provided in Table 9.
- The model parameters used to obtain Figure 10 & Table 3 and validate Principle 5 are provided in Table 10 and Table 11, respectively.
- The model parameters used to obtain Figure 11 and validate Principle 6 are provided in Table 12.

Table 6: Model parameters used for Figures 2, 6 & 7. These are the same parameters used in Poli et al. (2024).

| | |
|---|---|
| Coefficient Dynamics Parameters | Detailed in Section 5 |
| Key ($k_j$) & Querry ($q_i$) Preprocessing | None |
| Block Design | Transformer style (Type 1) |
| Number of Layers | 2 |
| Embedding Dimension $d$ | 128 |
| Inner (State) Dimension $n$ | 128 |
| Value Dimension $d_v$ | 128 |
| Number of Heads | 16 |
| Interleaved MLP Layers | Yes (dim $= 256$) |
| Positional Embedding | Yes |

Table 7: Model parameters used for Figures 8 & 9. These are the same parameters used in Poli et al. (2024).

| | |
|---|---|
| Coefficient Dynamics Parameters | Detailed in Appendix D.2 |
| Key ($k_j$) & Querry ($q_i$) Preprocessing | Detailed in Appendix D.2 |
| Block Design | Transformer style (Type 1) |
| Number of Layers | 2 |
| Embedding Dimension $d$ | 128 |
| Inner (State) Dimension $n$ | 128 |
| Value Dimension $d_v$ | 128 |
| Number of Heads | 16 |
| Interleaved MLP Layers | Yes (dim $= 256$) |
| Positional Embedding | Yes |

Table 8: Model parameters used for Figure 1. These are the same parameters used in Poli et al. (2024).

| | |
|---|---|
| Coefficient Dynamics Parameters | Detailed in Section 5 |
| Key ($k_j$) & Querry ($q_i$) Preprocessing | None |
| Block Design | Transformer style (Type 1) |
| Number of Layers | 2 |
| Embedding Dimension $d$ | 128 |
| Inner (State) Dimension $n$ | 128 |
| Value Dimension $d_v$ | 128 |
| Number of Heads | 16 |
| Interleaved MLP Layers | Yes (dim $= 256$) |
| Positional Embedding | Detailed in Section 5 |

Table 9: Model parameters used for Figure 3.

| | |
|---|---|
| Coefficient Dynamics Parameters | Detailed in Section 5 |
| Key ($k_j$) & Querry ($q_i$) Preprocessing | ShortConv(dim= 4) |
| Block Design | Gated DeltaNet style (Type 2) |
| Number of Layers | 2 |
| Embedding Dimension $d$ | 128 |
| Inner (State) Dimension $n$ | 128 |
| Value Dimension $d_v$ | 128 |
| Number of Heads | 8 |
| Interleaved MLP Layers | Yes (dim = 256) |
| Positional Embedding | No |

Table 10: Model parameters used for Figure 10. Below, if a parameter is declared with "or", the first option is for attention style models and the second for SSM/RNN style models.

| | |
|---|---|
| Coefficient Dynamics Parameters | Detailed in Appendix D.3 |
| Key ($k_j$) & Querry ($q_i$) Preprocessing | None or ShortConv(dim= 4) |
| Block Design | Type 1 or Type 2 |
| Number of Layers | 2 |
| Embedding Dimension $d$ | 128 |
| Inner (State) Dimension $n$ | $[128, 256, 512, 1024]$ |
| Value Dimension $d_v$ | 128 |
| Number of Heads | 4 |
| Interleaved MLP Layers | Yes (dim = 256) |
| Positional Embedding | Yes or No |

Table 11: Model parameters used for Table 3. Below, if a parameter is declared with "or", the first option is for softmax attention and the second for Mamba-2.

| | |
|---|---|
| Coefficient Dynamics Parameters | Softmax attention and Mamba-2 standard; varying $b_j$ |
| Key ($k_j$) & Querry ($q_i$) Preprocessing | None or ShortConv(dim= 4) |
| Block Design | Type 1 or Type 2 |
| Number of Layers | 8 |
| Embedding Dimension $d$ | 768 |
| Inner (State) Dimension $n$ | 768 |
| Value Dimension $d_v$ | 768 |
| Number of Heads | 12 |
| Interleaved MLP Layers | Yes (dim = 512) |
| Positional Embedding | Yes or No |

Table 12: Model parameters used for Figure 11.

| | |
|---|---|
| Coefficient Dynamics Parameters | Detailed in Appendix D.4 |
| Key ($k_j$) & Querry ($q_i$) Preprocessing | None |
| Block Design | Transformer style (Type 1) |
| Number of Layers | 2 |
| Embedding Dimension $d$ | 128 |
| Inner (State) Dimension $n$ | 128 |
| Value Dimension $d_v$ | 128 |
| Number of Heads | 16 |
| Interleaved MLP Layers | Yes (dim $= 256$) |
| Positional Embedding | Yes |

