# OpenReview forum: "Design Principles for Sequence Models via Coefficient Dynamics"
_ICLR.cc/2026/Conference — Submitted to ICLR 2026_

### Official Review · Reviewer_8c4e · 2025-10-16

**Soundness:** 2
**Presentation:** 3
**Contribution:** 1
**Rating:** 2
**Confidence:** 5

**Summary:**

This paper proposes a unified theoretical framework for viewing sequence models (including Transformers, recurrent neural networks (RNNs), and state-space models (SSMs)) through the lens of **coefficient dynamics**.
The authors model the coefficients alpha in the linear combination as outputs of an autonomous linear dynamical system driven by *impulse inputs*.
They claim this representation reveals shared mathematical structure among diverse sequence architectures, enabling the derivation of six **“design principles”** related to:

1. Linear vs. nonlinear readout maps and their efficiency tradeoffs
2. Input selectivity through the geometry of zero set
3. Encoding positional information via non-identity evolution matrices
4. Structured choices of A_t matrix (e.g., diagonal or Householder)
5. Proper scaling of injection parameters (b_j)
6. Normalization factors to ensure stability.

Experiments on synthetic tasks (MAD benchmark) empirically test these principles, showing expected patterns such as improved selectivity with larger zero sets and the sufficiency of (A_t that is not equal to I) to encode position without embeddings.

**Strengths:**

* Clear and mathematically consistent formulation.
* Provides a clean pedagogical summary connecting RNNs, attention, and SSMs under one algebraic form.
* Well-presented with readable equations and illustrative diagrams.
* Serves as a potential tutorial reference for newcomers to the field.

**Weaknesses:**

* **No novel theoretical result:** All lemmas rederive existing intuitions without advancing formal understanding.
* **Weak experimental validation:** Evaluations on simple synthetic tasks (MAD) do not test scalability, language modeling, or real-world data.
* **Limited empirical novelty:** Trends (e.g., gating helps, non-identity (A_t) adds positional info) are already widely known.
* **Incomplete discussion of prior work:** The paper fails to acknowledge prior theoretical frameworks that make similar connections. No point is not already known. I don't see anything in this paper that is novel and that we did not already know from training deep sequence models.

**Questions:**

1. **Scope of derivations:** Are any of your truly lemmas new, or are they reinterpretations of existing SSM analyses?
2. **Experimental depth:** Can you test your design principles on real-world data (e.g., language modeling or speech) to demonstrate their usefulness beyond toy MAD benchmarks?
3. **Relation to Mamba and Selective SSMs:** Since these models already embody your principles (learned (A_t,b_j), stability constraints), how does your framework offer new insights beyond theirs?
4. **Extension to multi-layer architectures:** The study is single-layer. How do the principles scale in multi-layer or cross-attention settings?

**Details Of Ethics Concerns:**

No concerns.

---

> ### Author Response · Authors · 2025-11-21
>
> Thank you for your in-depth review of our paper. We appreciate your time to go through the paper and your feedback. Thank you for your positive assessment of the presentation and the paper’s potential as a reference paper. We provide detailed discussions on your concerns below, due to the character limit we do this in two comments.
>
> **Weaknesses:**
>
> - Limited theoretical and empirical novelty: We agree that some of the design principles presented in the paper are already known and have been used or demonstrated by practitioners or experienced researchers in the field. We believe the novelty of the paper lies in the coefficient dynamics framework, which is the first framework to fully capture softmax attention and any linear attention proposal (e.g. Mamba, DeltaNet, linear RNNs, etc.) without resorting to approximations. Given this framework, it is possible to formalize existing practical knowledge in the design and applications of sequence models through mathematical lemmas. The goal of listing all of them, along with the novel ones (such as Principles 2 and 6) is to provide a comprehensive guideline that illustrates the impact of considering each of them in model design, and potential explanations for failure modes (such as insufficient scaling or normalization in Principles 5 and 6, respectively). In essence, the paper aims to identify the leitmotifs of modern sequence model design and provide a mathematical foundation to each of them. Finally, the framework also connects these principles, which allows to analyze the tradeoffs between the principles. Fundamentally, the main tradeoff for sequence models is between linear-in-time implementation (Principle 1) and expressivity formalized as the suppression of irrelevant information (Principle 2). These Principles are mutually exclusive, however there exist techniques to improve suppression (Principles 3, 4), which in turn might require targeted normalization or stabilization techniques (Principles 5, 6). We make sure to better highlight these arguments in the revised version.
> As an example of novelty consider Principle 2, which quantifies how the geometry of the zero-level set of the readout function affects the model’s expressivity (in practice this would be the near-zero-level set instead of the zero-level set). This e.g. provides a mathematical explanation for why linear attention models like Mamba-2 lack the efficient recall capabilities of softmax attention.
>
> - Weak experimental validation: We acknowledge that testing on language modeling would strengthen the paper. However, a synthetic benchmark like MAD has the benefit of testing targeted model capabilities that are needed for good language modeling. If we would directly test on a language task, these capabilities could not be tested individually. The MAD benchmark is explicitly designed to capture the exact capabilities needed for language modeling and good performance on the benchmark is indicative of good language model performance [1]. Additionally, we include an ablation on the Wikitext-103 dataset for Principle 5 in the appendix, as this Principle is best demonstrated with large dimensional keys, which otherwise is not informative on the MAD benchmark. Finally, to test all the principles in a fair and accurate manner, we would need to train several models in the billion parameter range, which is unfortunately out of your compute budget. In case you have a suggestion for a more realistic benchmark, which is not too computationally heavy, we would be happy to include additional experimental results in the revised version.
>
> - Incomplete discussion of prior work: To the best of our knowledge, we include the most relevant theoretical frameworks [2], [3], [4]. Additionally, we include the missing framework (MetaML, [5]) pointed out by other reviewers in the revised version. Are there any additional frameworks you would suggest to add? We are happy to include any missing references.
>
> **References:**
>
> [1] Poli et al. (2024), “Mechanistic Design and Scaling of Hybrid Architectures”, arXiv 2403.17844
>
> [2] Dao & Gu (2024), “Transformers are SSMs: Generalized Models and Efficient Algorithms Through Structured State Space Duality”, ICML
>
> [3] Yang et al. (2025), “PaTH Attention: Position Encoding via Accumulating Householder Transformations”, arXiv 2505.16381
>
> [4] Sieber et al. (2024), “Understanding the differences in Foundation Models: Attention, State Space Models, and Recurrent Neural Networks”, arXiv 2405.15731
>
> [5] Chou et al. (2024), “MetaLA: Unified Optimal Linear Approximation to Softmax Attention Map”, NeurIPS
>
> [6] Yang et al. (2024), “Gated Delta Networks: Improving Mamba2 with Delta Rule”, arXiv 2412.06464

---

> > ### Author Response · Authors · 2025-11-21
> >
> > **Questions:**
> >
> > 1. We believe all presented lemmas are novel in the sense that they provide a mathematical foundation to potentially already known empirical findings. While we agree that some principles might have been empirically validated in the literature (Principles 1, 3, 4, 5), the presented mathematical lemmas go deeper than the principles themselves. For example, Principle 3 (positional embedding) foundationally connects, the well-known fact that linear attention methods like Mamba do not need positional embeddings with methods like ALiBi that essentially introduce an $A \neq I$ to softmax attention. Such a connection and its effect is novel to the best of our judgement. We believe that Principles 2 and 6 (and their corresponding lemmas) are truly new since they both capture phenomena not directly rooted in well-known empirical findings. Principle 2 explains how nonlinear attention methods like softmax attention are better at suppressing irrelevant information than linear attention methods and Principle 6 explains why a method like xLSTM without a stable A can work. Both of which add insights to methods in the literature.
> > 2. While ablations on the MAD benchmark provide some predictive empirical evidence on the performance of language models [1], we agree that ablations on a realistic language modeling task would strengthen the results of the paper. However, testing on a language model task would require us to train several models in the billion parameter range, which is unfortunately not in our compute budget. Additionally, the MAD benchmark allows us to specifically test targeted capabilities, which are needed for language modeling.
> > 3. We would like to point out that the framework not only captures linear RNN models like Mamba, DeltaNet, etc. but also softmax attention and its variants. Therefore, these principles also hold for softmax attention. In that, the framework provides direct insights across the complete spectrum of sequence models and does not need to resort to approximations like in SSD [2]. For instance, the scaling factor $b_j$ in Mamba-2 is originally derived from discretization and only happens to align with Principle 5 (see Example 2 in the paper). Additionally, our framework also predicts that the parameters A and b in Mamba-2 do not need to be coupled, which has been partially acknowledged in Gated DeltaNet [6], albeit inexplicitly.
> > 4. We acknowledge that the single layer analysis of sequence models is one of the limitations of this paper. However, going beyond a single layer adds a tremendous amount of complexity, which is out of scope for this paper. We plan to analyze this case by virtue of cascading systems from dynamical systems theory in future work.
> >
> > **References:**
> >
> > [1] Poli et al. (2024), “Mechanistic Design and Scaling of Hybrid Architectures”, arXiv 2403.17844
> >
> > [2] Dao & Gu (2024), “Transformers are SSMs: Generalized Models and Efficient Algorithms Through Structured State Space Duality”, ICML
> >
> > [3] Yang et al. (2025), “PaTH Attention: Position Encoding via Accumulating Householder Transformations”, arXiv 2505.16381
> >
> > [4] Sieber et al. (2024), “Understanding the differences in Foundation Models: Attention, State Space Models, and Recurrent Neural Networks”, arXiv 2405.15731
> >
> > [5] Chou et al. (2024), “MetaLA: Unified Optimal Linear Approximation to Softmax Attention Map”, NeurIPS
> >
> > [6] Yang et al. (2024), “Gated Delta Networks: Improving Mamba2 with Delta Rule”, arXiv 2412.06464

---

> ### Comment · Reviewer_8c4e · 2025-11-24
> **Open to raise score in case experiments and detailed conclusions on BabyLM are done**
>
> Since you explicitly ask for a "more realistic but not too computationally heavy" benchmark, a natural compromise would be to add a small-scale language-modeling experiment on a BabyLM-style corpus, e.g., the Strict-Small track (10M words). BabyLM is explicitly designed for low-compute LM research and comes with an off-the-shelf evaluation pipeline (perplexity + BLiMP, SuperGLUE-style tasks, etc.), while staying in the 10M–100M token regime.
>
> Concretely, training one moderately sized version of your architecture (e.g. 50–100M parameters) on the 10M-word BabyLM corpus, and comparing variants that include vs. exclude a given principle (e.g. Principle 2 or 6) would already go a long way towards addressing my concern: it would show that the "coefficient dynamics" design choices matter for actual language modeling, not just for MAD synthetics, but without requiring billion-parameter models or massive datasets.
>
> If these experiments are conducted, I am happy to raise my score.

---

> > ### Author Response · Authors · 2025-11-24
> >
> > Dear Reviewer
> >
> > Thank you very much for suggesting the BabyLM benchmark. We were not aware of this benchmark and we are happy to include as many ablations on this benchmark as time permits. We agree that results on this benchmark would bolster the claims of the paper.
> >
> > We already started integrating the benchmark in our code base and will come back to you as soon as we have first results. In the mean time, do you have additional feedback on the other points raised in the review/rebuttal?
> >
> > Thank you for the constructive feedback and for considering to raise your score.

---

> > > ### Comment · Reviewer_8c4e · 2025-11-28
> > > **Waiting for BabyLM experiments and conclusions.**
> > >
> > > No additional points.

---

### Official Review · Reviewer_gmof · 2025-10-31

**Soundness:** 2
**Presentation:** 3
**Contribution:** 2
**Rating:** 4
**Confidence:** 3

**Summary:**

This paper proposes a number of principles for designing sequence modeling layers. To come up with such proposals, the paper provides a survey of previous works, and mechanistically separate parts into A, b, \phi, \eta, and \alpha, highlighting major different parts. Coming up with six principles, the authors provide ablations for each, validating their claims.

**Strengths:**

The paper is well structured, proposing several ideas that not only help designing new architectures but also understand existing methods. The principles that are introduced in the paper are simple, thus can be easily adopted. Each proposal is backed up with an experiment.

**Weaknesses:**

My major concern of this paper is that the ideas are only validated using synthetic benchmarks as these benchmarks cannot model all real-world issues, and quite vulnerable to training setups such as weight decay and learning rates. Since the paper comes up with the principles that each should improve the model, I believe the authors should have used some real-world datasets (say language modeling) and benchmark at least their best model that all the principles are applied.

Additionally, I also have concerns with some principles. For example, Principle 1 and 5 are quite trivial which are already well known among the community, and Principle 6 is not specific enough (what specifically is an unstable A?). Also, I wonder if Principle 3 is actually correct: for instance, attention-based autoregressive models without positional embeddings (i.e., NoPE) has shown promising results.

**Questions:**

- What would be the model with all the principles applied, and how does it perform on language modeling?

- Is Principle 3 true?

---

> ### Author Response · Authors · 2025-11-21
>
> Thank you for your in-depth review of our paper. We appreciate your time to go through the paper and your feedback. Thank you for your positive assessment of the framework and the simplicity of the principles. We provide detailed discussions on your concerns below.
>
> **Weaknesses:**
>
> We agree that ablations on a more realistic task like language modeling would strengthen the paper. However, a synthetic benchmark like MAD has the benefit of testing targeted model capabilities that are needed for good language modeling (e.g. recall, memorization, compression). If we would directly test on a language task, these capabilities could not be tested. This is also the reasoning behind many other works in the area that primarily test on synthetic benchmarks (e.g. [1]). The MAD benchmark is explicitly designed to capture the exact capabilities needed for language modeling and good performance on the benchmark is indicative of good language model performance [2]. Additionally, we would need to train several models in the billion parameter range to accurately test these capabilities at scale, which is unfortunately out of our compute budget. Finally, we include an ablation on Wikitext-103 for Principle 5 in the appendix.
>
> We agree that Principles 1 & 5 might be well-known in the existing literature and be experienced practitioners. However, the lemmas underlying these principles also offer more fundamental insights given their mathematical grounding. For instance, Principle 5 stipulates that we need to scale the keys appropriately, which for the case of Mamba is not necessarily fulfilled and only arises from the normalization with the timestep Delta. Additionally, these principles hold for both linear and nonlinear sequence model designs, which allows for direct comparisons between methods, e.g., the discounting effect of A in Mamba or DeltaNet and the positional embedding method ALiBi for softmax attention.
>
> Thank you for pointing out the missing definition of an unstable $A$ matrix. We include the following formal definition and a remark in the revised paper.
>
> **Definition:** Matrix $A \in \mathbb R^n$ is said to be stable iff all eigenvalues of $A$, i.e. $\lambda(A)$, lie inside the unit circle of the complex plane, i.e., $|\lambda_i(A)| \le 1 \forall i = 1, ... ,n$. Conversely, matrix $A$ is said to be unstable iff any of the eigenvalues $\lambda(A)$ lies outside the unit circle of the complex plane, i.e., $\exists i = 1, …, n, s.t. |\lambda_i(A)| \ge 1$.
>
> Regarding Principle 3, it is mathematically correct, see Appendix C.1.3 in the paper. However, there exist tasks for which positional embeddings are less important, explaining good performance of softmax attention with NoPE. Please also see our answer to your Question 2 for more details.
>
> **Questions:**
>
> 1. The optimal sequence model would be task & requirement dependent, i.e., according to the principles, there is no single best sequence model in that sense. Fundamentally, the paper highlights the tradeoff between linear-in-time implementation (Principle 1) and expressivity formalized as the suppression of irrelevant information (Principle 2). These Principles are mutually exclusive. However, there exist techniques to improve suppression (Principles 3, 4), which in turn may require targeted normalization or stabilization techniques (Principles 5, 6). In the revised version, we make this point clearer. Regarding the application to language modeling: the optimal sequence model according to the principles depends on your preference of speed (linear-in-time implementation) and expressivity. In either case, you base the design on linear attention (e.g. Mamba) or a Transformer, respectively.
> 2. Yes, Principle 3 holds in general. The fact that linear attention models ($A \neq I$) work well without positional embeddings has been discussed in e.g. [3], while the effectiveness of methods like RoPE and AliBi can also be explained by Principle 3. For some applications, the performance might not depend on accurately distinguishing between identical inputs that are processed at different time instances. However, for a task like recall this is essential [4]. A very good treatment of this topic for hybrid models (both nonlinear and linear attention layers) has been recently given in Kimi Linear [5], where the authors show that NoPE for the nonlinear layers is possible because of the interleaved linear layers with $A \neq I$.
>
> **References:**
>
> [1] Merrill et al. (2024), “The Illusion of State in State-Space Models”, arXiv 2404.08819
>
> [2] Poli et al. (2024), “Mechanistic Design and Scaling of Hybrid Architectures”, arXiv 2403.17844
>
> [3] Dao & Gu (2024), “Transformers are SSMs: Generalized Models and Efficient Algorithms Through Structured State Space Duality”, ICML
>
> [4] Arora et al. (2023), “Zoology: Measuring and Improving Recall in Efficient Language Models”, arXiv 2312.04927
>
> [5] Kimi Team (2025), “Kimi Linear: An Expressive, Efficient Attention Architecture”, arXiv 2510.26692

---

### Official Review · Reviewer_gPgH · 2025-11-01

**Soundness:** 3
**Presentation:** 2
**Contribution:** 2
**Rating:** 6
**Confidence:** 2

**Summary:**

This paper proposes a unified theoretical framework to describe sequence models including Transformer and RNNs, and puts forward a series of design principles and experimental verifications to guide sequence modeling.

**Strengths:**

1. The design principles for sequence modeling architecture proposed in this paper are instructive.
2. The analysis combining theory and experiment is convincing.

**Weaknesses:**

A unified sequence modeling framework for transformer and RNN model architectures has been mentioned in various works, such as MetaLA [1], PaTH Attention [2], and log-linear-attention [3], which may diminish the contribution of this paper. Therefore, further comparison and discussion with similar related works will help highlight the contribution of this paper.

[1] Yuhong Chou, et al. MetaLA: Unified Optimal Linear Approximation to Softmax Attention Map. NeurIPS, 2024.
[2] Songlin Yang, et al. PaTH Attention: Position Encoding via Accumulating Householder Transformations. NeurIPS, 2025.
[3] Han Guo, et al. Log-Linear Attention. arXiv, 2025.

**Questions:**

1. Based on the design principles of the sequence modeling architecture proposed in the paper, what characteristics should the optimal model architecture for sequence modeling?
2. Although these design principles can guide the design of the optimal sequence modeling architecture, for linear RNN architectures, the design of some components may not be conducive to hardware efficient parallel training of RNN. In this case, would a more general and expressive matrix gate be less useful than a simpler, more efficiently trained diagonal/scaler gate?

---

> ### Author Response · Authors · 2025-11-21
>
> Thank you for your in-depth review of our paper. We appreciate your time to go through the paper and your feedback. Thank you for the assessment that the paper is insightful. We provide detailed discussions on your concerns below.
>
> **Weaknesses:**
>
> We believe the novelty of the paper lies in the coefficient dynamics framework, which is the first framework to fully capture softmax attention and any linear attention proposal (e.g. Mamba, DeltaNet, linear RNNs, etc.) without resorting to approximations. As the review correctly points out, there exist frameworks like PaTH, log-linear attention, etc. that attempt to explain softmax attention and linear attention models. However, these frameworks work on a different level than what our framework does. E.g. log-linear attention is a practical method to extend the constant memory of e.g. Mamba (this would be the compressed history of every KV pair seen) to growing memory to better mimic the behaviour of the KV cache in softmax attention. Our framework on the other hand captures the fundamental dynamics of a sequence layer like softmax attention or linear attention or Mamba or DeltaNet. This is also different from e.g. MetaLA, which only captures linear attention models. In the revised version of the paper, we make this distinction clearer and also include MetaLA [1], which we did not cite in the original version.
>
> **Questions:**
>
> 1. The optimal sequence model would be task dependent, i..e, according to the principles there is no single best sequence model in that sense. Fundamentally, the paper highlights the tradeoff between linear-in-time implementation (Principle 1) and expressivity formalized as the suppression of irrelevant information (Principle 2). These Principles are mutually exclusive, however there exist techniques to improve suppression (Principles 3, 4), which in turn might require targeted normalization or stabilization techniques (Principles 5, 6), although these also hold in general. In the revised version, we make this point clearer.
> 2. Thank you for bringing this up. In the revised version we add a remark on this point. We believe that a full discussion on efficient implementation is out of scope for this paper. There already exist methods that e.g. attempt to improve expressivity of Mamba by allowing negative eigenvalues [2], yet are not implementable with current kernels. Therefore, the actual fast and efficient implementation of any of the presented design principles or their extensions is a separate issue. Additionally, we would like to point out that these principles also hold for softmax attention and its variants.
>
> **References:**
>
> [1] Chou et al. (2024), “MetaLA: Unified Optimal Linear Approximation to Softmax Attention Map”, NeurIPS
>
> [2] Grazzi et al. (2024), “Unlocking State-Tracking in Linear RNNs Through Negative Eigenvalues”, arXiv 2411.12537

---

### Official Review · Reviewer_SX4M · 2025-11-04

**Soundness:** 3
**Presentation:** 3
**Contribution:** 2
**Rating:** 4
**Confidence:** 5

**Summary:**

1. The paper proposes a unified theoretical framework (Coefficient Dynamics) for analyzing sequence models, including Transformers, linear attention, and State Space Models (SSMs). The framework formalizes the fact that these models compute outputs as linear combinations of past tokens, and interprets the outputs of *linear dynamical systems driven by impulse inputs*.

2. Unlike prior unification approaches, this formulation explicitly introduces a per-token index, j, representing each previous key/value, conceptually similar to a KV-cache, and shows Transformers, SSMs, Linear Attention-like architectures can be expressed as special cases.


Building on this framework, the authors derive a set of design principles:

  1. **Linearity of $\phi$:** Only linear readout maps permit parallel recurrent computation.
  2. **Input selectivity:** Nonlinear $\phi$ enables sparse coefficients aka selectivity.
  3. **Positional information:** Setting $A_t \neq I$ embeds position into coefficients; $A_t = I$ requires positional embeddings.
  4. **Evolution structure:** The choice of $A_t$ (scalar, diagonal, Householder) allows for transformations such as scaling or rotation.
  5. **Scaling of $b_j$:** Proper scaling of $b_j$, prevents variance blow-up with increasing hidden state.
  6. **Normalization $\eta_i$:** Stable training requires normalization of coefficient magnitudes.

**Strengths:**

1. The paper is clearly written and the mathematical formulation is rigorous.
2. Unifies existing insights on SSMs, RNNs, and attention into a single framework summarizing core principles like linearity, efficiency, input selectivity, normalization, and stability. This is pedagogically useful for newcomers to the field.

**Weaknesses:**

### **On the framework**

The “coefficient dynamics” construction, builds on standard frameworks like Dynamical Systems Framework with the new ingredient being the explicit *per token j index*, which is equivalent to maintaining a key–value (KV) cache. In my opinion this viewpoint is not novel in a theoretical sense: prior work (e.g., Dao & Gu, 2024; Sieber et al., 2024) already expresses attention and SSMs as linear recurrences or matrix multiplications over past states. The main change, vis-a-vis previous works, is the KV cache-like formulation to unify attention without using infinite state sizes.

---

### **On Principle 1 ($\phi$ must be linear for parallelization)**

In my opinion, this is a well-known result in the subquadratic. Specifically, it is known that only linear readouts permit associative-scan like formulations required for efficient recurrent computation. This principle has been used in multiple works (Linear Attention, Mamba, Mamba-2, Gated Deltanet). While the lemma is correct, its inclusion as a “new principle” adds little beyond reiterating that *linear functions yield linear complexity*.

---

### **On Principle 2 (Input selectivity and geometry of $\phi$)**

The main idea—that nonlinear ϕ enables suppression of uninformative tokens while linear ϕ limits selectivity—is sound and nicely presented. However, this observation is well known and connects to classic results on **associative memory capacity**, where linear associative memories can store only ~n patterns in dimension n.

As a nit remark: the follow-up discussion "Can learnable parameters save us?" is technically correct but does not logically follow from the principle, since modifying $A_t$ or $b_j$ changes only the key–value dynamics, not the query-dependent coefficients $\alpha_{ij}$.

Remark: Authors claim that Linear Attention has a readout of the form $\psi(\cdot)\psi(\cdot)$, but this is not really a function acting post the readout as defined and hence does not fit with the framework definition. It is better viewed as a preprocessing trick rather than a part of the framework.

---

### **On Principle 3 (Positional Information)**

The result that $A_t = I$, which implies that per-query token the sequence mixing process is permutation invariant and hence requires position embeddings is also well known and has been discussed in prior works. Authors of Mamba mention that due to the decay, the operation is no longer permutation invariant and hence does not require position embeddings. In my opinion, the lemma correctly states this but adds no new insight.

---

### **On Principle 4 (Structure of $A_t$)**

The fact that the structure of the state transition matrix $A_t$ (scalar, diagonal, Householder) limits the operation that can be performed on the keys is a tautological statement for me as the state transition matrix is what acts on the keys to produce the output. In the Lemma, authors simple summarize the actions performed by scalar, diagonal, or Householder matrix operators on the keys being acted upon.

---

### **On Principles 5 & 6 (Scaling and normalization)**

The discussion on $b_j = O(1/\sqrt{n})$ to maintain $O(1)$ variance and the normalization of coefficients $\alpha_{ij}$ to avoid exploding norms repeats standard initialization theory (Glorot & Bengio, 2010; Vaswani et al., 2017). While correctly stated, these are rules of thumbs which are widely used in architecture design to ensure that variance remains bounded in deep models. In my opinion, their inclusion as novel “principles” is overstated.

---

My overall assessment of novelty is that the paper’s strength lies in gathering well-known rules of thumbs under a shared formalism, but every individual principle has been studied or applied before.

**Questions:**

N/A

---

> ### Author Response · Authors · 2025-11-21
>
> Thank you for your in-depth review of our paper. We appreciate your time to go through all the math and your feedback on the individual principles. Below we detail our response to your concerns, due to the character limit, we do this in two separate comments.
>
>
> **Weaknesses:**
>
> We would like to start by addressing what we believe is the main contribution of our work. The main novelty introduced by the paper lies in the coefficient dynamics framework, which is the first framework to fully capture softmax attention and any linear attention proposal (e.g. Mamba, DeltaNet, linear RNNs, etc.) without resorting to approximations. As pointed out in the review, this does relate to the KV cache idea, however this is much more general, as the KV cache for a linear attention architecture like Mamba would be its hidden state, i.e., the compressed history of all KV pairs seen. Given this framework, it is possible to formalize existing practical knowledge in the design and applications of sequence models through mathematical lemmas. While we absolutely agree that some of the design principles presented in the paper are already known and have been used or demonstrated by practitioners or experienced researchers in the field, this paper is the first to unify them all in a common framework and provide mathematical grounding to each of them. The goal of listing all of them, along with the novel ones (such as Principles 2 and 6) is to provide a comprehensive guideline that illustrates the impact of considering each of them in model design, and potential explanations for failure modes (such as insufficient scaling or normalization in Principles 5 and 6, respectively). In essence, the paper aims to identify the leitmotifs of modern sequence model design and provide a mathematical foundation to each of them. To that end, we would like to highlight the important role of the lemmas introduced in the paper as a novel mathematical formalization for the principles. We will modify the paper to appropriately highlight the contributions and clarify where some of the principles listed have been used before.
>
> An example of an important insight derived from the coefficient dynamics framework is the identification of a fundamental trade-off between linear-in-time implementation (Principle 1) and expressivity (Principle 2 & subpriciples) with potential tweaks to improve expressivity and stability given other design choices (remaining principles). We believe this framework allows easy comparison between sequence models (crucially including softmax attention) and identifies trade-offs that are not immediately obvious from a given model proposal. For instance, power retention [1] that uses $\phi = (.)^2$, i.e., a nonlinear readout function but which can be implemented in a linear form using an exact kernel function. At first glance, this proposes a true nonlinear readout map but with linear implementation. However, this readout map suffers from the same limitations as an identity (or linear) readout with respect to Principle 2, as it is not better at suppressing coefficients. Such an architecture proposal can be immediately analyzed and compared to other architectures with our framework.
>
> **References:**
>
> [1] Gelada et al. (2025), “Scaling Context Requires Rethinking Attention”, arXiv 2507.04239

---

> > ### Author Response · Authors · 2025-11-21
> >
> > **Principles:**
> >
> > - Principles 1, 4, 5: We agree that these are empirically well-known to practitioners in the field and are included here to illustrate the mathematical insights derived from the framework for the sake of completeness. We also point out that, despite the widespread knowledge of these principles, the mathematical formalization provides a more nuanced view, e.g., Principle 5 stipulates that we need to scale the keys appropriately, which for the case of Mamba is not necessarily fulfilled and only arises from the normalization with the timestep Delta. Additionally, these principles connect the underlying reasoning behind the design of nonlinear and linear sequence models and the arising trade-offs. In the revised version, we make sure to highlight this better.
> >
> > - Principle 2: We agree that theory on associate memory capacity overlaps partially with our Subprinciple 2.2, however we believe the formalization and the math behind Subprinciple 2.2 and the corresponding lemma are novel. We highlight and additionally remark on this in the revised version. However, we would like to point out that Principle 2 itself goes much further than this, since it quantifies how the geometry of the zero-level set of the readout function affects the models expressivity (in practice this would be the near-zero-level set instead of the zero-level set). This e.g. provides a mathematical explanation for why linear attention models like Mamba-2 lack the efficient recall capabilities of softmax attention.
> >
> > - Principle 3: The review correctly points out that Mamba and subsequent linear attention proposals do not need positional embeddings due to their permutation invariance. However, the principle also captures newer positional embedding methods like ALiBi that essentially incorporate an $A \neq \mathbb{I}$ in softmax attention and softmax attention variants like forget attention. Therefore, the principle allows direct comparisons between all these proposals and provides a unified explanation of why these work.
> >
> > - Principle 6: We believe this principle to be novel in that it explains why a method like xLSTM works. Except for xLSTM, we are not aware of any other sequence model that allows dynamics matrices A with eigenvalues outside the unit circle, as such a method without additional normalization $\eta$ will blow-up over longer sequences. Additionally, this principle gives rise to interesting geometric interpretations of sequence model outputs as laid out in Appendix A (although we are not the first ones to point this out).

---

### Author Response · Authors · 2025-12-03
**Summary for new Area Chair**

Dear Area Chair,

We are writing to summarize the current state of our submission following the reset of the review process. As you are new to our paper, our goal is to provide a concise overview of reviewer feedback and the key clarifications and revisions we have made.

Our paper introduces the coefficient dynamics framework, the first framework that exactly captures the dynamics of softmax attention and most linear attention models (e.g., Mamba, DeltaNet, linear RNNs) without approximations. This framework unifies architectures, grounds widely used design heuristics in mathematical lemmas, and reveals a fundamental trade-off between linear-time computation and expressivity defined as the suppression of irrelevant information. Reviewers consistently recognized the promise and usefulness of the framework, but criticized the novelty. We believe this criticism is mainly due to the paper’s failure to clearly state the contributions, which we rectified in the rebuttal.

**Key Improvements and Clarifications**
- *Clearer positioning of novelty:* We now explicitly state the novelty of the paper by strengthening the discussion of the coefficient dynamics framework and positioning it as the first framework to capture softmax attention without approximations.
- *Better theoretical clarity:* We expanded explanations of the fundamental trade-off between linear-time implementation (Principle 1) and expressivity (Principle 2), and added a formal definition of stability for dynamics matrices relevant to Principle 6.
- *Improved discussion of principles:* We clarified which principles reflect known empirical behavior and which (Principles 2 and 6) introduce genuinely new insights. For all principles, we highlight that the mathematical foundation is novel and provide further details on Principles 2 & 6, which add mathematical explanations for phenomena such as softmax attention’s recall ability and the viability of architectures with unstable dynamics (e.g., xLSTM).
- *Expanded related work:* We added the missing MetaLA framework and confirmed that all other relevant theoretical perspectives are included.
- *Clarification of Empirical experiments:* We clarified the role and motivation of the MAD benchmark as a targeted test for model capabilities and added a Wikitext-103 ablation for Principle 5.

Below is a reviewer-by-reviewer summary of feedback and our response:

**Reviewer SX4M**
- Insights: Questioned the novelty of some principles and asked for clarification of how our contributions extend beyond well-known heuristics.
- Addressed by: Highlighting that the framework’s novelty lies in (1) the unification of all these principles under a single mathematical formalism, (2) new lemmas that provide previously missing theoretical grounding, and (3) genuinely new principles (e.g., suppression geometry and stability with unstable dynamics; Principles 2 & 6). Clarified trade-offs and strengthened explanations of expressivity limitations in linear attention models.
- Result: No reply before review roll-back.

**Reviewer gPgH**
- Insights: Asked about novelty relative to PaTH, log-linear attention, and MetaLA, and about how the principles inform the best sequence model.
- Addressed by: Providing clear distinctions between our framework and prior work, explaining that ours captures fundamental dynamics applicable to both softmax and linear models. Clarified that no single architecture is optimal: models face a fundamental trade-off between linear-in-time computation and expressive suppression, with additional principles governing how to compensate. Therefore, the optimal model is task and objective dependent.
- Result: No reply before review roll-back.

**Reviewer gmof**
- Insights: Requested more realistic empirical validation, clarification of stability definitions, and justification of principles known in practice.
- Addressed by: Explaining the purpose and advantages of the MAD benchmark, adding a formal definition of stable vs. unstable dynamics matrices, and clarifying that while some principles reflect known empirical phenomena, our lemmas provide novel theoretical explanations applicable across both linear and nonlinear attention models.
- Result: No reply before review roll-back.

**Reviewer 8c4e**
- Insights: Raised concerns about theoretical novelty, experimental scope, and related work completeness.
- Addressed by: Emphasizing the novelty of the coefficient dynamics framework, especially Principles 2 & 6; expanding discussion of trade-offs across principles; explaining compute constraints and adding Wikitext-103 ablations; updating related work with MetaLA and verifying completeness.
Result: The reviewer was open to raise the score if we include ablations on the BabyLM benchmark.

We believe these revisions and clarifications significantly strengthen our paper, addressing all substantive concerns, while making the contributions clearer and more rigorous. Thank you for your time and consideration.

Sincerely,
Authors

---

### Meta-Review · Area_Chair_9zT1 · 2026-01-10

**Summary:**

The paper proposes a unified framework to analyze deep sequence models, including Transformers (Softmax Attention), State Space Models (SSMs), and gated linear RNNs, by modeling their output coefficients as autonomous linear dynamical systems driven by impulse inputs. The work aims to derive fundamental design principles by identifying tradeoffs between expressivity, implementation efficiency, and stability conditions, attempting to provide a guiding theoretical compass for future sequence model architectures.

On the other hand, the paper have the following drawbacks:
- Lack of Substantive Novelty and Insight:
While the attempt to build a general framework is commendable and intellectually interesting, the conceptual novelty is limited. The framework overlaps significantly with existing perspectives that connect linear attention and linear RNNs. More importantly, the "design principles" derived do not lead to meaningful new insights. For instance, the framework does not yield a novel implementation that improves computational complexity or a new architecture that outperforms existing baselines. The lemmas presented are largely reformulations of well-known properties in the literature, failing to push the boundary of our current understanding.
- Insufficient Empirical Validation: The experimental section is a primary weakness. The analysis is mostly theoretical or illustrated with small-scale examples. To justify the claim of providing "guiding principles for systematically designing new architectures," the paper needs to demonstrate these principles on large-scale, real-world datasets and competitive benchmarks.

The AC recognizes the elegance of unifying various models under a single dynamical systems viewpoint. However, in its current form, the paper functions more as a review or a re-categorization of known facts rather than a contribution of new knowledge or practical tools. Due to the lack of novel insights and the absence of robust experiments on real-world data, the paper does not go beyond the bar for acceptance at ICLR.

**Reviewer Concerns:**

- Although Reviewer gPgH provided a short review, their comment is appropriate and thus it is not of concern.
- The author raised a concern that Reviewer 8c4e's comment is vague and is suspected as AI generated one. Their suggestion to conducting experiments on BabyLM looks a bit out of point, however, their comments are not as pointless as what is blamed for AI generation.

**Reviewer Scores:**

- The score by Reviewer 8c4e could have been updated to score 4. However, the total evaluation by the reviewers would not have met the ICLR acceptance bar.

---

### Decision · Program_Chairs · 2026-01-26

Reject